



Natural Hazards
and Earth System
Sciences

# Hydrometeorological droughts in the Miño–Limia–Sil hydrographic demarcation (northwestern Iberian Peninsula): the role of atmospheric drivers

**Rogert Sorí**[1], **Marta Vázquez**[1], **Milica Stojanovic**[2], **Raquel Nieto**[1], **Margarida L. R. Liberado**[2,3], **and Luis Gimeno**[1]

[1]Environmental Physics Laboratory (EPhysLab), CIM-UVigo, Universidade de Vigo, 32004 Ourense, Spain
[2]Instituto Dom Luiz, Faculdade de Ciências, CE1 Universidade de Lisboa, 1749-016 Campo Grande, Portugal
[3]Escola de Ciências e Tecnologia, Universidade de Trás-os-Montes e Alto Douro, 5001-801 Vila Real, Portugal

**Correspondence:** Rogert Sorí (rogert.sori@uvigo.es)

**Abstract.** TS1 CE2 Drought is one of the world's primary natural hazards because of its environmental, economic, and social impacts. Therefore, monitoring and prediction for small regions, countries, or whole continents are challenging. In this work, the meteorological droughts affecting the Miño–Limia–Sil hydrographic demarcation in the northwestern Iberian Peninsula during the period of 1980–2017 were identified. For this purpose and to assess the combined effects of temperature and precipitation on drought conditions, the 1-month standardized precipitation evapotranspiration index (SPEI1) was utilized. Some of the most severe episodes occurred during June 2016–January 2017, September 2011–March 2012, and December 2014–August 2015. An empirical-orthogonal-function analysis revealed that the spatial variability of the SPEI1 shows strong homogeneity in the region, and the drought phenomenon consequently behaves in the same way. Particular emphasis was given to investigating atmospheric circulation as a driver of different drought conditions. To this aim, a daily weather type classification based on the Lamb weather type (LWT) classification was utilized for the entire Iberian Peninsula. Results showed that atmospheric circulation from the southwest, west, and northwest are directly related to wet conditions in the Miño–Limia–Sil hydrographic demarcation during the entire hydrological year. Contrastingly, weather types imposing atmospheric circulation from the northeast, east, and southeast are best associated with dry conditions. Anomalies of the integrated vertical flow of humidity and their divergence for the onset, peak, and termination of the 10 most severe drought episodes also confirmed these results. In this sense, the major atmospheric teleconnection patterns related to dry and wet conditions were the Arctic Oscillation, Scandinavian pattern, and North Atlantic Oscillation. Hydrological drought investigated through the standardized runoff index was closely related to dry and wet conditions revealed by the SPEI at shorter temporal scales (1–2 months), especially during the rainy months (December–April).

## 1 Introduction

Drought is one of the most dangerous natural phenomena in many regions worldwide, as it affects a wide range of environmental, economic, and social sectors (Wilhite, 2000; McMichael and Lindgren, 2011 TS2; Stanke et al., 2013; Gerber and Mirzabaev, 2017; Liberato et al., 2017; Guerreiro et al., 2018). This phenomenon is usually considered to be a prolonged dry period in the natural hydrologic cycle that can occur anywhere in the world. It is initially caused by a lack of rainfall as well as by thermodynamic processes (e.g. turbulent fluxes and water phase transitions; Wehrli et al., 2018) induced by aerodynamics (wind speed), radiation forces (solar and long-wave), and thermal forces (high temperatures) (Vicente-Serrano et al., 2010; Seneviratne, 2012; WMO and GWP, 2016; Miralles et al., 2019). Drought propagation is also due to natural and human drivers through multiple feedbacks (Van Loon et al., 2016). The Iberian Peninsula (IP) in the Euro-Atlantic and Mediterranean regions is a drought-

prone area (Páscoa et al., 2017) that was affected by severe droughts in 2004–2005 (García-Herrera et al., 2007), 2011–2012 (Trigo et al., 2013), and 2016–2017 (García-Herrera et al., 2019). Concerning the existence of the trends in droughts, Lloyd-Hughes and Saunders (2002) found a significant negative linear trend on the series of the Palmer Drought Severity Index (PDSI) over the period of 1901–1999 in the northwestern IP (hereafter NWIP). Higher atmospheric evaporative demand increased the severity of climatic droughts during the period of 1961–2011 in the IP, which contributed to a decrease in surface water resources (Vicente-Serrano et al., 2014). However, these authors also argued that drought variability has mainly been controlled by precipitation. Coll et al. (2017) reported that the rise in temperature was responsible for the greater drought severity and larger surface area affected in the IP from the 1980s to 2010 (with respect to the period of 1906–2010), as it led to an increase in atmospheric evaporative demand. A significant tendency towards dryness during 1975–2012 in the IP was also revealed by Páscoa et al. (2017). These authors showed that the northwestern region of the IP was particularly affected by these trends. Over a shorter study period (1974–2010), Gómez-Gesteira et al. (2011) found a significant increasing trend of 0.5 °C per decade in air temperature in this same region and 0.24 °C per decade in the sea surface temperatures of the adjacent Atlantic Ocean, but annual precipitation did not show any significant trend in the interior, which suggested a possible dominant role of temperature on the occurrence of drought in this region. Although the results described above agree with respect to the occurrence of a trend towards drier conditions in the IP during recent decades, the findings of Spinioni et al. (2017) reveal noticeable differences with respect to the trends and severity of droughts among the winters of 1950–2014 and 1981–2014 and the spring, summer, and autumn seasons of 1950–2015 and 1981–2015 in the IP. Overall, the high confidence level that global warming is likely to reach 1.5 °C above preindustrial levels within a short period (between 2030 and 2052) if the current rate of increase continues is presently a serious concern (IPCC, 2018). In this sense, the IP is considered to be one of the most likely European regions to suffer from an increase in drought severity during the 21st century (Spinioni et al., 2018). However, Trenberth et al. (2014) argued that increased heating from global warming may not necessarily cause more droughts, but when they do occur, they would be expected to exhibit a rapid onset and greater intensity.

Drought processes involve interactions amongst ocean processes (ocean teleconnections), land-based processes (water balance and runoff), and several atmospheric processes (Spinioni et al., 2017). Therefore, multiple analyses have been used to investigate droughts and their impact on the availability of water resources in the IP. These include the implementation of circulation weather type (CWT) classifications (Cortesi et al., 2014; Ramos et al., 2014); identification of atmospheric blocking events (Sousa et al.,

2016); and assessments of climatic teleconnection patterns like the North Atlantic Oscillation (NAO; Muñoz-Díaz and Rodrigo, 2004; Trigo et al., 2004; deCastro et al., 2006a), which is considered to be a dominant mode of climate variability for Europe (Visbeck et al., 2001); the Arctic Oscillation (AO; deCastro et al., 2006a); the El Niño–Southern Oscillation (ENSO) (Vicente-Serrano, 2005); and the Scandinavian pattern (SCAND; deCastro et al., 2006a). The impacts of drought in the IP have been widely investigated. Droughts have been found to affect the productivity of rainfed crops (Peña-Gallardo et al., 2019) and forests (Gouveia et al., 2009; Barbeta and Peñuelas, 2016; Vidal-Macua et al., 2017; Peña-Gallardo et al., 2019) and have even resulted in human mortality in Galicia, northwestern Spain (Salvador et al., 2019). Terrestrial ecosystems often vary significantly in their responses to drought (Knapp et al., 2015). Moreover, the IP is characterized by different climate types, which vary from a humid Atlantic climate in the northwest and north to a semi-arid Mediterranean climate in the east and southeast (Parracho et al., 2016); it also features strong seasonal variability (Serrano et al., 1999). Therefore, regional-scale studies would have the advantage of better characterizing the phenomenon of drought and its impacts in this region, thereby supporting the reduction of drought vulnerability as well as drought-induced losses.

The NWIP is a hydrologically important region where water resources of the Miño–Sil and Limia River basins represent an important source of benefits for inhabitants of Galicia and the northern provinces of Portugal. Both basins make up the Miño–Limia–Sil hydrographic demarcation (MLSHD; Fig. 1), a region of shared environmental resources for Spain and Portugal. The water resources in the MLSHD are crucial to developing agriculture and livestock. Indeed, in the Spanish portion of the MLSHD, 73.2 % of the total water demand is for agrarian use (Vargas and Paneque, 2019), principally irrigation (PH, 2014). The industrial use of water for energy production is mainly carried out through hydroelectric and thermal power plants, which has caused hydromorphological alterations in the primary river channels due to the construction of dams and dikes (deCastro et al., 2006b; PH, 2014). The hydroelectric power plants in the Spanish part of the MLSHD have a percentage of installed power that supposes a gross production of 5878.18 GWh (average in the period of 2004–2013), which represents 44.88 % of the total generated in the region (PH, 2014).

To the best of our knowledge, there are few published studies about drought considering the MLSHD as a whole. Ojeda et al. (2019) investigated the temporal evolution of agricultural and hydrological droughts in the major river basins of the IP during the period of 1980–2014 through modelling datasets. Considering the importance of the MLSHD as a hydrological unit, our main aim is to not only investigate the meteorological droughts which have affected the MLSHD using high-resolution gridded datasets but also to evaluate the role of atmospheric circulation and large-

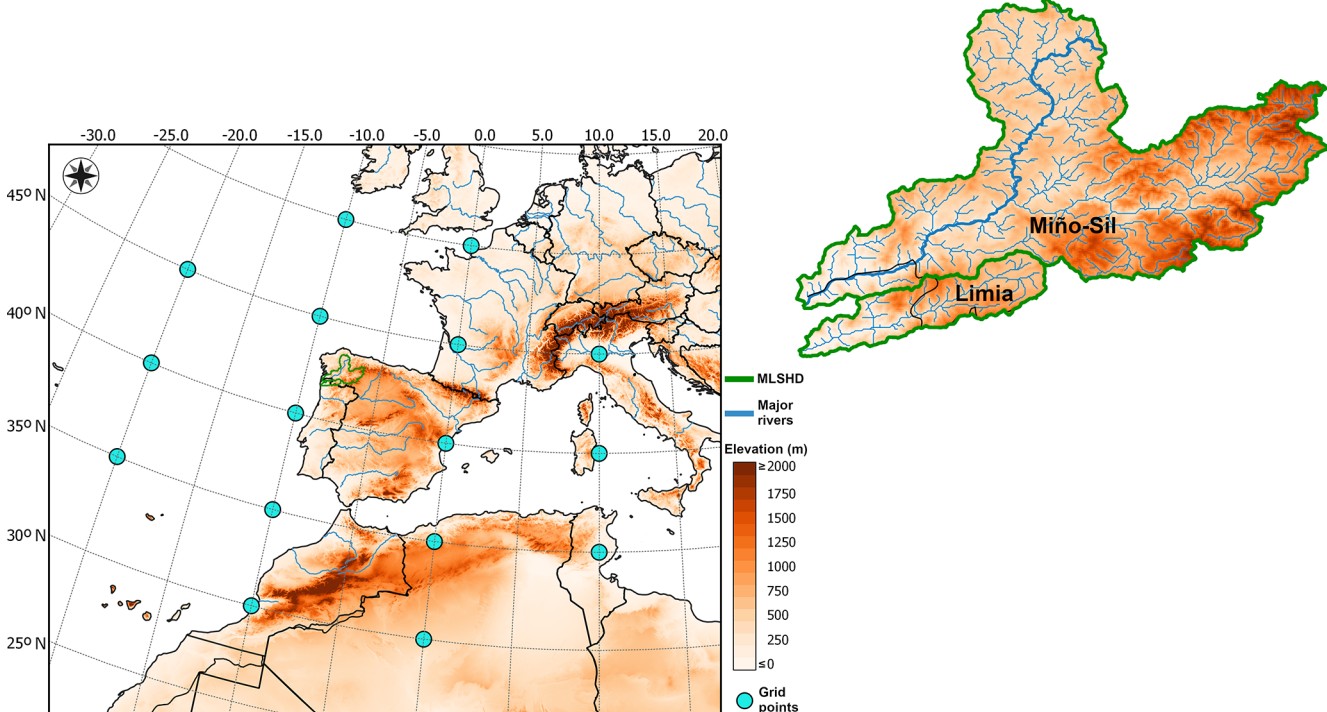

**Figure 1.** Geographic location and boundaries (green line) of the Miño–Sil and Limia River basins which conform to the Miño–Limia–Sil hydrographic demarcation (MLSHD). The rivers are represented by blue lines, and elevation is shaded in red (in metres above sea level) from the HydroSHEDS project (Hydrological data and maps based on SHuttle Elevation Derivatives at multiple Scales; CE3 Lehner et al., 2011). Light-blue circles denote the location of the 16 points used to retrieve daily mean sea level pressure (MSLP) values for the computation of the circulation weather types (CWTs).

scale teleconnection patterns in the occurrence and magnitude of droughts over a longer period (1980–2017). Russo et al. (2015) carried out a similar analysis for the entire IP, but their approach identified the seasonal conditions of the drought state and related them to the frequency of weather types during 1950–2012. In this study, we focus on meteorological droughts for explaining the primary cause of other types of droughts (e.g. agricultural, hydrological, and socioeconomic). These are normally associated with the reduction of the soil moisture content; low river flows; low water levels in rivers, lakes, and groundwater; and socioeconomic impacts due to water scarcity (WMO, 2012). Hydrological and agricultural droughts are primarily driven by meteorological droughts; therefore, their forecasts also heavily depend on weather forecasting. Identifying mechanisms that drive meteorological droughts and investigating drought characteristics is essential to improve forecasting methods and proactively monitor for early warnings, which contribute to efficient water management and the preservation of the ecosystems and socioeconomic stability. We expect that our results will contribute to increasing the hydroclimate knowledge of the MLSHD, support early-warning forecasting, and strengthen drought management plans.

## 1.1 Study area

The MLSHD extends from approximately 42 to 44° N and from 6.5 to 9° W and covers an area of approximately 20 000 km$^2$ in the NWIP (Fig. 1), including the territories of Galicia (Spain) and northern Portugal. There are 191 municipalities, of which 181 belong to the Spanish territory and 10 to the Portuguese territory, with a total population of 1 084 636 people (as of 2015; Mora-Aliseda et al., 2015). It is designated as a "management unit", where the terrestrial area is composed of the Miño–Sil and Limia River basins and the transitional, subterranean, and coastal waters are associated with basins (PES, 2017). The MLSHD is characterized by the presence of a diverse landscape based on a complex relief structure and Atlantic bioclimatic characteristics. The Miño–Sil basins have a pronounced mountainous character with an average elevation of $\sim$ 683 m a.s.l. (above sea level; UN, 2011), while the Lima River basin is at $\sim$ 447 m a.s.l. CE4 (CA, 2020). The rugged coastlines, valleys, and mountains present a wide variety of landscapes in a unique combination compared to those of the surrounding peninsular territories. In terms of the annual mean flow, the Miño–Sil River basin is the fourth largest basin in the IP and because of that is important for hydropower generation (Lorenzo-Lacruz et al., 2013; Añel et al., 2014).

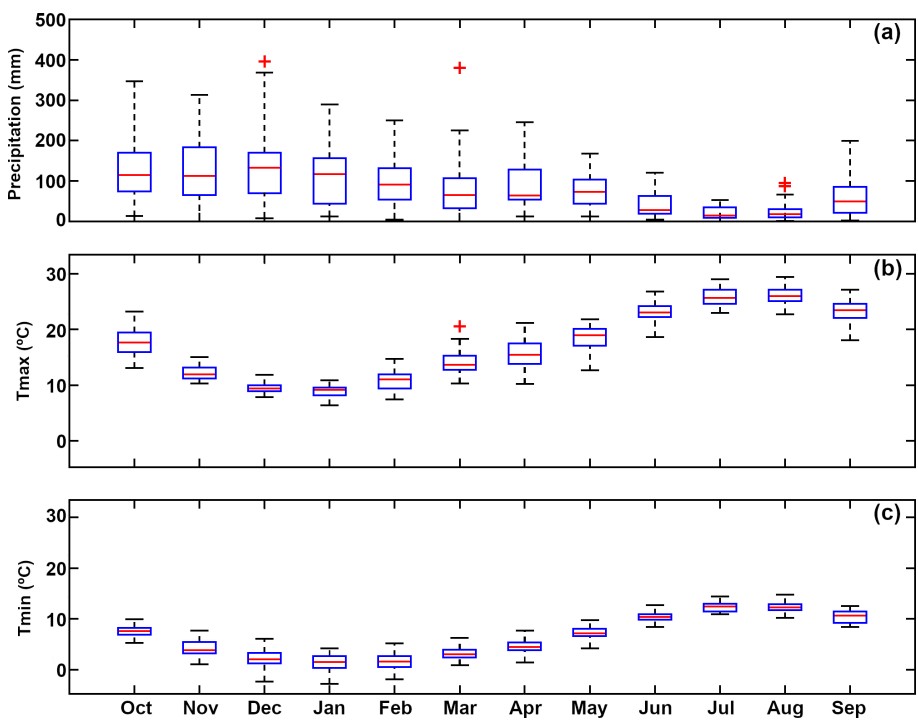

**Figure 2.** The annual cycle for the hydrological year spanning October of year $n$ to September of year $n+1$, for the period of 1980–2017, for **(a)** monthly mean precipitation, **(b)** maximum temperature, and **(c)** minimum temperature in the MLSHD, using the E-OBS gridded dataset. Boxes delineate the median (red line), upper, and lower quartiles, with the whiskers representing the lowest and highest monthly value still within 1.5 of the interquartile range. Outliers, i.e. values beyond the ends of the whiskers, are denoted by a + symbol.

Its climate is characterized by mild winters, cool summers, humid air, abundant cloudiness, and frequent rainfall. A box plot of the monthly mean precipitation, maximum temperature, and minimum temperature from the E-OBS CE5 gridded dataset (Cornes et al., 2018) reveals the hydrological year (October of year $n$ to September of year $n+1$) in the MLSHD (Fig. 2). The precipitation presents a high temporal variability across the year. According to the values of the median, the rainiest months are December and January, while the less rainy months are July and August (Fig. 2a). As expected, the greatest scattering for precipitation occurs during the autumn and winter months. Both the Miño and the Sil are remarkably regulated rivers, although they have a maximum flow in winter (January and February) and a minimum in summer (August and September; Añel et al., 2014). During winter, the large-scale circulation is mainly driven by the position and intensity of the Iceland Low, and western Iberia is affected by westerly winds that bring humid air and generate precipitation (Trigo et al., 2004). The movement of the sub-tropical anticyclone to the south leaves the region open to the influence of the frontal systems from the west, which are responsible for most of the precipitation. Synoptic-scale baroclinic perturbations from the Atlantic Ocean are responsible for most of the precipitation between October and May (deCastro et al., 2006a). Indeed, this period of months can be considered as the rainy season. Summer is predominantly in-

fluenced by high pressures, which determine air subsidence and consequently atmospheric stability (PGRH, 2016 TS3). The annual cycle of maximum and minimum temperature reveals an opposite cycle to that of precipitation (Fig. 2b and c). According to the quartile distribution of monthly mean temperatures values, the coldest months are December, January, and February; in these months, the extreme values of the average minimum temperature for the MLSHD has dropped to values below 0 °C. Finally, according to the interquartile range distribution of the maximum average temperature and middle values of 25.6 and 25.9 °C, July and August, respectively, are the hottest months.

## 2    Material and methods

### 2.1    Datasets

Monthly gridded data of precipitation, maximum temperature, and minimum temperature were obtained from daily values of the E-OBS v.18e gridded dataset (Cornes et al., 2018) with a longitudinal and latitudinal resolution of 0.1° for the period of 1980–2017. Owing to their high resolution, these datasets have been utilized to investigate extreme precipitation (Tabari and Willems, 2018) and drought events in Europe (Manning et al., 2019). However, the sparse distribution network in some European regions has led to an over-

smoothing of precipitation intensities (Hofstra et al., 2009, 2010; Sunyer et al., 2013; Herrera et al., 2019). A comparison between daily precipitation and temperatures from standard and ensemble E-OBS datasets with the observational gridded dataset Iberia01 performed by Herrera et al. (2019) revealed that the main differences of temperatures occurred in the south, around the Guadalquivir and Guadiana basins, and respect the precipitation the high biases in the central IP and the Mediterranean regions. In addition to the high resolution, these datasets were chosen for this study because they provide both precipitation and temperature fields necessary for the computation of the standardized precipitation evapotranspiration index (SPEI), thus minimizing errors that could arise due to the mixing of different datasets.

The period of 1980–2017 was set for all the analyses in this study taking into account the simultaneous availability of data and a period of more than 30 years. These datasets were utilized to compute the SPEI in the MLSHD. For the CWT computation, daily values of sea level pressure (SLP) from the ERA-Interim (ECMWF Reanalysis) reanalysis datasets (Dee et al., 2001 TS4) with a resolution of 1°, based on the 16 grid points shown in Fig. 1, were utilized. The eastward–northward vertically integrated moisture flux from ERA-Interim was utilized to compute the anomalies of vertical integral moisture flux (VIMF) and its divergence anomalies. Monthly values of runoff with a resolution of $\sim 4$ km were freely downloaded from the portal TerraClimate (available at http://www.climatologylab.org/terraclimate.html, TS5; Abatzoglou et al., 2018). TerraClimate uses climatically aided interpolation, combining high-spatial-resolution climatological normals from the WorldClim dataset with coarser-spatial-resolution and time-varying data from CRU TS4.0 (Climate Research Unit Time Series) and the Japanese 55-year Reanalysis (JRA55).

To identify the influence of short and large-scale modes of climate variability on the hydroclimate of the study region, various datasets of teleconnection patterns were used. These were the bivariate ENSO time series (BEST; available at https://www.esrl.noaa.gov/psd/people/cathy.smith/best/, TS6; Smith and Sardeshmukh, 2000), the Western Mediterranean Oscillation (WeMO; available at https://crudata.uea.ac.uk/cru/data/moi/, TS7), East Atlantic (EA) CE6, NAO, AO, and SCAND (available at https://www.cpc.ncep.noaa.gov/data/teledoc/telecontents. shtml, TS8). The BEST index time series is based on the combination of the atmospheric component of the ENSO phenomenon (the Southern Oscillation Index; SOI) and an oceanic component (average Niño 3.4 sea surface temperature). The WeMO index (WeMOi) is based on the difference between the standardized atmospheric pressure recorded at Padua (45.40° N, 11.48° E) in northern Italy and San Fernando Cádiz (36.28° N, 6.12° W) in southwestern Spain. To obtain the teleconnection indices of the Northern Hemisphere (EA, NAO, AO, and SCAND), the Climate Prediction Center (CPC) applies a rotated principal component analysis (RPCA) proposed by Barnston and Livezey (1987), using the monthly mean standardized 500 mb height anomalies from the NCEP/NCAR Reanalysis (National Centers for Environmental Prediction; National Center for Atmospheric Research; Climate Data Assimilation System – CDAS) in the analysis region of 20–90° N.

## 2.2 Drought identification: the standardized precipitation evapotranspiration index (SPEI)

Different types of droughts make it difficult to conceive a universal drought index. Therefore, there are many indices and different criteria to identify and investigate different types of droughts (Svodoba and Fuchs, 2016; WMO and GWP, 2016 TS9). However, ultimately drought is caused by an imbalance between water supply and demand. Therefore, the SPEI (Vicente-Serrano et al., 2010) was chosen to identify dry conditions in the MLSHD during the period of 1980–2017. This index is based on the same methodology of the standardized precipitation index (SPI; McKee et al., 1993). However, as an advantage over common precipitation-based drought indices (e.g. the SPI), the SPEI considers the effects of temperature through the reference parameter of evapotranspiration ($Et_0$ TS10) in the climatic water balance represented in Eq. (1):

$$D = (P - Et_0),\qquad(1)$$

where $D$ is the water balance over a given period of time, $P$ is the precipitation that represents water availability, and $Et_0$ represents the atmospheric water demand. Therefore, the SPEI combines the changes the atmospheric evaporative demand with the multiscalar nature of the SPI (Beguería et al., 2014), which allows for the assessment of the response of different ecological, hydrological, and agricultural systems to drought (Vicente-Serrano et al., 2012). In consequence, it has been applied to a large variety of ecosystems across the world for identifying dry and wet conditions and evaluating drought recurrence (Potop et al., 2013; Vicente-Serrano et al., 2016; Salah et al., 2018 TS11; Sordo-Ward et al., 2017; Wang et al., 2018). It was also chosen for this study because the results of Vicente-Serrano et al. (2014) described how drought severity has increased in the past 6 decades (1954–2014) in natural, regulated, and highly regulated basins of the IP as a consequence of greater atmospheric evaporative demand resulting from temperature rise.

$Et_0$ values are a climatic parameter that expresses the evaporating power of the atmosphere at a specific location and time of the year (Allen et al., 1998). In the absence of meteorological data required for applying the Penman–Monteith equation, which is recommended by the Food and Agriculture Organization (FAO) of the United Nations in the FAO bulletin 56 (Allen et al., 1998), we used the method proposed by Hargreaves and Samani (1985). It is based on temperature data to estimate the $Et_0$ value according to Eq. (2):

$$Et_0 = 0.408 \cdot Ch \cdot Ra \cdot \left(\sqrt{T_x - T_n}\right) + (T_m + 17.8),\qquad(2)$$

Please note the remarks at the end of the manuscript.

where CE7 Ch $= 0.0023$; Ra is the extra-terrestrial radiation (derived from the latitude and the month of the year); and $T_x$ TS12, $T_n$ TS13, and $T_m$ TS14 are the maximum, minimum, and mean temperature, respectively. By this method, we do not consider relative humidity and wind speed to also be important factors for determining the vapour density above the soil surface and the aerodynamic resistance for vapour transport, which permit more realistic $Et_0$ values and consequently better drought assessment (Bittelli et al., 2008; Vicente-Serrano et al., 2010; WMO, 2012; Davarzani et al., 2014). However, even though the Penman–Monteith equation offers a more accurate estimation of reference $Et_0$ values than the Hargreaves formula (López-Moreno et al., 2009; Tomas-Burguera et al., 2017), results of Vicente-Serrano et al. (2014) showed that $Et_0$ values in Spain estimated by the Hargreaves–Samani method for the period of 1961–2011 had the closest agreement with the $Et_0$ values obtained by the Penman–Monteith method in terms of temporal evolution and magnitude with respect to over 11 other methods. These authors also found high correlations between an $Et_0$ value obtained by both methods in the NWIP.

The resultant $D$ values in Eq. (1) were aggregated at different timescales, following the same procedure as the SPI. According to Vicente-Serrano et al. (2010), Beguería et al. (2014), and Vicente-Serrano and Beguería (2016), the developers of the index, to calculate the SPEI at different timescales, the most suitable statistical distribution to model the $D$ series is the log-logistic distribution, which is given by Eq. (3):

$$F(D) = \left[ 1 + \left( \frac{\alpha}{D - \gamma} \right)^{\beta} \right]^{-1}, \tag{3}$$

where $\alpha$, $\beta$, and $\gamma$ represent the scale, shape, and location parameters that are estimated from the sample $D$. Finally, the SPEI is obtained as the standardized values of $F(D)$. Previous studies have been also used the log-logistic distribution to obtain the SPEI series for the IP (e.g. Russo et al., 2015; Páscoa et al., 2017; Coll et al., 2017; Ojeda et al., 2019). For our study region, it is possible that the $D$ values demonstrate a better fit to a different probabilistic distribution; however, this can also occur for different accumulation periods of $D$ (Monish and Rehana, 2020). The use of different probabilistic distributions to fit the $D$ series may primarily affect the tail of each distribution and the extreme SPEI values (Vicente-Serrano et al., 2010; Vicente-Serrano and Beguería, 2016), e.g. the $[-2.33, 2.33]$ bounds (one event in 100 cases). For these reasons, we preferred to use the distribution suggested by the authors of the index to calculate the SPEI on a timescale of 1 to 24 months. For the calculation of the SPEI, the R package available at http://cran.r-project.org/web/packages/SPEI ( TS15 ) is utilized. It includes all the recommendations proposed by Beguería et al. (2014).

**Table 1.** SPEI classification according to Agnew (2000).

| SPEI | Probability | Category |
|---|---|---|
| $> 1.65$ | 0.05 | Extremely humid |
| $> 1.28$ | 0.10 | Severely humid |
| $> 0.84$ | 0.20 | Moderately humid |
| $> -0.84$ and $< 0.84$ | 0.60 | Normal |
| $< -0.84$ | 0.20 | Moderately dry |
| $< -1.28$ | 0.10 | Severely dry |
| $< -1.65$ | 0.05 | Extremely dry |

We avoided considering that any precipitation below the mean constitutes a drought. Therefore, the classification of drought categories for SPI values proposed by Agnew (2000) (Table 1) was utilized in this study. Other authors have also employed this classification for investigating drought in the IP (e.g. Vicente-Serrano, 2005 TS16 ; Páscoa et al., 2017). This classification, despite being pre-established, was built by probability classes rather than magnitudes of the SPI and is, therefore, a more rational approach, with a noticeable effect at the demarcation of mild and moderate droughts (Agnew, 2000). We focus on the SPEI at a 1-month temporal scale (hereafter SPEI1) to identify meteorological drought episodes. At this timescale the SPEI and the SPI can reflect short-term conditions, and consequently, its application can be closely related to meteorological types of droughts (WMO, 2012). A drought episode occurs every time the SPEI1 is continuously negative and reaches the value of $-0.84$ or less. The onset of an episode is the month in which the episode begins; the peak is the month in which the episodes reach the highest negative value of the SPEI1; and the end is the last month that the SPEI1 is negative. The threshold of $-0.84$ corresponds to 20 % probabilities, whereby a drought is expected to occur once in 5 years, which reduces the incidence of mild meteorological droughts. The duration is calculated as the sum of the consecutive number of months with negative SPEI values since the onset of the episode, and the severity is calculated as the sum of all SPEI values (in absolute values) during the episode.

To identify the principal patterns of drought variability in the MLSHD, an empirical-orthogonal-function (EOF) analysis (Preisendorfer and Mobley, 1988; Von Storch and Navarra, 1995 TS17 ) was utilized. The EOF analysis is not based on physical principles; rather, the technique aims to decompose observed datasets into two components that capture most of the observed variance in space (eigenvalues) and time (eigenvectors). It makes easier to study the principal modes of variability of the SPEI1 time series for every grid point of the MLSHD. The percentage of the total variance explained by each eigenvalue is able to explain most of the spatial drought variance. This method has been extensively used to investigate droughts at global (e.g. Dai, 2011) and regional (Wang et al., 2017, 2019) scales.

Trend analysis using the Mann–Kendall test (Mann, 1945; Kendall, 1975) of pre-whitened time series data in the presence of serial correlation using the approach of Von Storch and Navarra (1995 TS18) was performed. This approach ensures the avoidance of possible autocorrelation of the series. The null hypothesis ($H_0$ TS19) is that the data come from a population with independent realizations and are identically distributed, while the alternative hypothesis ($H_a$ TS20), is that the data follow a monotonic trend. With a significance level of 0.05, the null hypothesis of no trend is rejected if the $|Z|$ score is greater than the critical value 1.96. Sen's slope (Sen, 1968) was used to determine the magnitude of trend increasing or decreasing in the period of study. The combination of both methods has often been used to analyse the trend change of hydrometeorological time series data such as precipitation, runoff, and the drought index. For this analysis, the R package "modifiedmk" (Patakamuri and O'Brien, 2019) was used.

## 2.3 The standardized runoff index (SRI)

The SRI was applied to investigate the occurrence and temporal evolution of hydrological droughts in the MLSHD. To compute this index, we used the same approach employed by McKee et al. (1993) to compute the SPI. According to these authors, the procedure can be applied to other variables relevant to drought, e.g. streamflow or reservoir contents. Thus, the gamma distribution was used for fitting monthly runoff data for accumulation periods up to 6 months.

## 2.4 Weather type classification methodology

Synoptic systems are linked to the dominant climate in any region of the planet. These systems represent the general circulation of the atmosphere through different configurations of variables. For this reason, a semi-objective classification scheme based on the methodology adopted by Trigo and Da-Camara (2000) from the Jenkinson and Collison (1977) and Jones et al. (1993) circulation schemes is applied to obtain the dominant circulation weather types (CWTs) over the IP. The method uses daily mean sea level pressure (MSLP) values obtained from the ERA-Interim reanalysis for the period of 1980–2017 on 16 different points over the IP and surrounding regions (light-blue circles are shown in Fig. 1) to build a set of indices associated with the direction and vorticity of the geostrophic flow. These are the total shear vorticity ($Z$), southerly shear vorticity ($Z_S$ TS21), westerly shear vorticity ($Z_W$ TS22), total flow ($F$), southerly flow ($S_F$ TS23), and westerly flow ($W_F$ TS24). The area used to compute CWTs was the same as that used by Ramos et al. (2014); it extended from 20° W to 10° E longitudes and from 30 to 50° N latitudes. The regional indices were computed as follows; Eqs. (4) to (9) were calculated according to the procedure described in Ramos et al. (2014) and Trigo and DaCamara (2000).

$$S_F = 1.305 \left[ 0.25 \left( p_5 + 2 \times p_9 + p_{13} \right) \right.$$
$$\left. -0.25 \left( p_4 + 2 \times p_8 + p_{12} \right) \right] \tag{4}$$

$$W_F = \left[ 0.5 \left( p_{12} + p_{13} \right) - 0.5 \left( p_4 + p_5 \right) \right] \tag{5}$$

$$Z_S = 0.85 \times \left[ 0.25 \left( p_6 + 2 \times p_{10} + p_{14} \right) \right.$$
$$-0.25 \left( p_5 + 2 \times p_9 + p_{13} \right)$$
$$-0.25 \times \left( p_4 + 2 \times p_8 + p_{12} \right)$$
$$\left. +0.25 \left( p_3 + 2 \times p_7 + p_{11} \right) \right] \tag{6}$$

$$Z_W = 1.12 \times \left[ 0.5 \times \left( p_{15} + p_{16} \right) - 0.5 \times \left( p_8 + p_9 \right) \right]$$
$$-0.91 \times \left[ 0.5 \times \left( p_8 + p_9 \right) - 0.5 \times \left( p_1 + p_2 \right) \right] \tag{7}$$

$$F = \left( S_F^2 + W_F^2 \right)^{1/2} \tag{8}$$

$$Z = Z_S + Z_W \tag{9}$$

Following this approach, 26 CWTs were initially identified (10 pure, 8 anticyclonic hybrids, and 8 cyclonic hybrids). The 16 points designated $p_x$ TS25 ($x$ going from 1 to 16 according to the number of the point represented in Fig. 1) were taken into account in this computation. Purely directional CWTs – northeastern (NE), eastern (E), southeastern (SE), northwestern (NW), western (W), southwestern (SW), north (N), and south (S) – were those showing $|Z| < F$ with the direction defined by $\tan -1 \left( W_F / S_F \right)$ (180° was added if $W_F$ was positive). If $|Z| > 2F$, then the circulation would be considered cyclonic (C) (if $Z > 0$) or anticyclonic (A) (if $Z < 0$). As not all the circulation patterns could be associated with a pure (directional, cyclonic, or anticyclonic) type, 16 hybrid circulations were defined as a combination of A and C circulations with directional CWTs. In this case, it is $F < |Z| < 2F$. Following Trigo and DaCamara (2000) in the frequency computation, the 26 CWTs were regrouped in the 10 pure circulations. Thus, each of the 16 hybrid types counted equally as a half occurrence to each of their corresponding purely directional and cyclonic or anticyclonic types (e.g. one case of CNW was included as 0.5 in C and 0.5 in NW). This same methodology but with a different source of mean sea level pressure datasets and lower resolution has been previously applied to investigate the relationships between atmospheric circulation and precipitation variability (e.g. Cortesi et al., 2014; Ramos et al., 2014) or drought conditions in the IP (e.g. Russo et al., 2015).

## 2.5 Weather type classification methodology

Wavelet coherence (WC) analysis was used to identify which frequency bands within two time series were co-varying (Torrence and Webster, 1999). This definition is similar to that of a traditional cross correlation, and the WC was considered as a localized correlation coefficient in time–frequency space (Torrence and Compo, 1998; Grinsted et al., 2004). For this assessment, the SPEI1 as well as the 6-month temporal-scale series of teleconnection patterns, namely the

BEST, WeMO, NAO, AO, EA, and SCAND patterns, were utilized by initially applying Eq. (10), as follows:

$$R_n^2(S) = \frac{\left| S\left(s^{-1} W_n^{XY}(s)\right)\right|^2}{S\left(s^{-1}\left|W_n^X(s)\right|^2\right) \cdot S\left(s^{-1}\left|W_n^Y(s)\right|^2\right)} \quad (10)$$

where $S$ is a smoothing operator and the variable $XY$ is the two series. The WC ranged from 0 to 1; if the value was closer to 1, then the correlation between the two series was higher. The cross-wavelet coherence analysis was performed using the "wtc" function through the "biwavelet" R package (https://CRAN.R-project.org/package=biwavelet, TS26), and the WC 5 % significance level was determined using 1000 randomizations of Monte Carlo-generated noise. An advantage of the WC over the classical cross-correlation analysis is that the phase relationship is calculated such that the degree to which two time series are positively or negatively related can be measured as both a function of time and period (Schulte et al., 2016).

## 3 Results and discussion

### 3.1 Drought conditions

The NWIP is a homogeneous region in terms of the total precipitation variance over the IP (Rodriguez-Puebla et al., 1998; Muñoz-Díaz and Rodrigo, 2004) and consequently, also in terms of the influence of droughts (Russo et al., 2015). Figure 3 illustrates the temporal evolution of moderate, severely, and extremely dry conditions (according to the classification shown in Table 1) through the SPEI at a temporal scale of 1 to 24 months. As observed, dry conditions revealed by the short-term SPEI were more frequent and propagated at larger SPEI temporal scales. This shows a lagging and lengthening effect in the drought signal, from meteorological to hydrological droughts (Wang et al., 2016; Gu et al., 2020). A visual analysis confirms that after 2004 the MLSHD has been more frequently affected by severe droughts.

In order to investigate meteorological droughts in the MLSHD, an EOF analysis was performed with the gridded values of the SPEI1. The results in Fig. 4 reveal the spatial characteristics of the first three leading EOF modes, which explain 97.8 % of the total spatial variability of the SPEI1 in the MLSHD and the corresponding PCs CE8. Because the spatial patterns of the remaining EOFs explain very low percentages, they are not shown. In particular, the EOF1 is characterized by entirely negative values, indicating that dry or wet conditions in the MLSHD manifest themselves homogeneously throughout its entire extension in a great percentage (93 %). The spatial coefficients of the EOF2 separate the eastern highlands from the north and west of the MLSHD, while the EOF3 reveals the north–south spatial differences. However, both explain just 2.6 % and 2.2 %, respectively. The PC1 exhibits less temporal variability than

**Table 2.** Trend analysis according to the pre-whitened McDonald–Kreitman (MK) CE9 tests and Sen's slope for the 1-month SPEI series and the number of episodes that start per year, their duration, and their severity. Statistically significant trends at ($p < 0.05$) and $Z$ scores are marked with an asterisk. Period of 1980–2017.

|                    | $Z$    | Slope     | Units              |
|--------------------|--------|-----------|--------------------|
| SPEI1              | −1.98  | −0.0007*  | $Z$ unit per year  |
| Number of episodes | 0.40   | 0.0030    | Episodes per year  |
| Duration           | 1.66   | 0.0200    | Months per year    |
| Severity           | 1.97   | 0.0300*   | $Z$ unit per year  |

the PC2 and PC3, and none of the three presents statistically significant trends.

Figure 5a shows the temporal evolution of the SPEI1 computed for the MLSHD during 1980–2017. Drought conditions are observed in periods such as 1989–1992, 2004–2007, and 2015–2017, in agreement with results obtained by other authors for the NWIP through different indices (e.g. García-Herrera et al., 2007 TS27; Andrade and Belo-Pereira, 2015 TS28; Spinioni et al., 2016; Ojeda et al., 2019). At this timescale, the negative values of the SPEI were primarily related to meteorological drought, which was unable to diagnose the agricultural, hydrological, and socioeconomic types of drought that are typically associated with the SPEI at greater temporal scales (WMO, 2012). However, meteorological droughts can be perceived as the initial cause of further types of droughts, since these are triggered in this case by the deficit of precipitation combined with high temperatures and significant $Et_0$ values. The identification of meteorological drought episodes affecting the IP has been a topic of research during recent years (e.g. Lana et al., 2006; Lorenzo-Lacruz et al., 2013; Páscoa et al., 2017; González-Hidalgo et al., 2018). A drought episode was considered to occur every time the SPEI1 was continuously negative and reached the value of −0.84 or less; this threshold was identified by the black dashed line in Fig. 5a. The onset, termination, and duration of these episodes are shown in Fig. 5b.

A tendency analysis for our period of the study reveals a small negative trend in the series of the SPEI1 (statistically significant), an increase in the number and duration of the episodes (statistically non-significant), and a small increase in the severity of these episodes (statistically significant) (Table 2). Vicente-Serrano et al. (2011) also found that the mean duration of drought episodes in the NWIP increased by approximately 1 month in the last 30 years of 1930–2006 (difference statistically non-significant) as a consequence of the increase in a potential $Et_0$ value.

Extreme drought events can disrupt food production systems and thus be a significant natural trigger for famine (Wilhite, 2000), and for the MLSHD they can directly affect hydroelectric production. The top 10 driest episodes in the period under this study according to their severity are shown in Table 3. This selection was made to develop further analy-

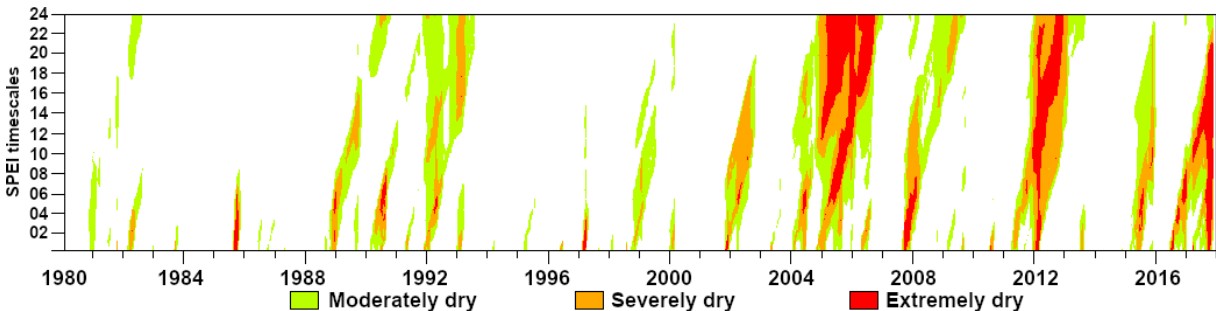

**Figure 3.** Temporal evolution of moderately, severely, and extremely dry conditions in the MLSHD according to the SPEI at a temporal scale of 1 to 24 months and the classification shown in Table 1. Period of 1980–2017.

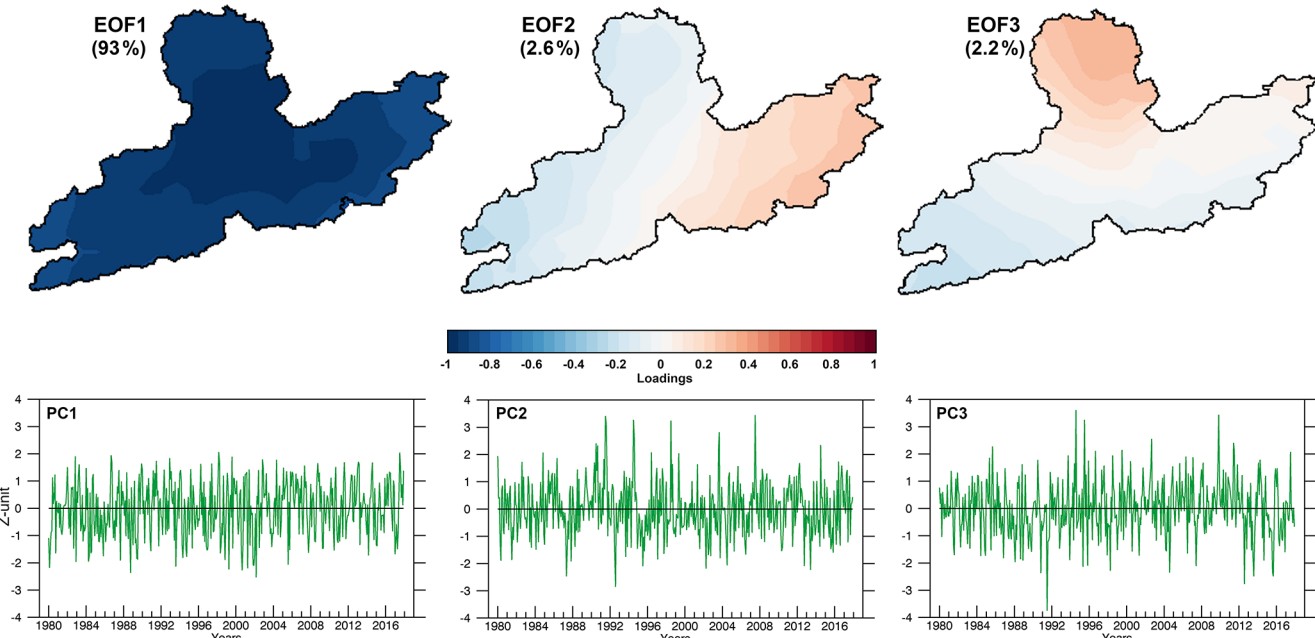

**Figure 4.** The first three leading EOF modes of the SPEI1 for the MLSHD. Period of 1980–2017. The numbers between parentheses correspond to the amount of variance explained by each EOF.

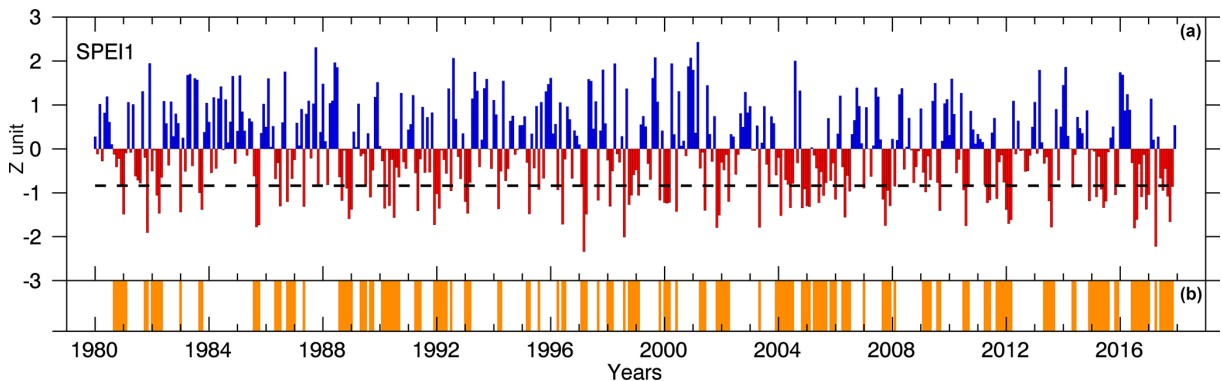

**Figure 5.** Wet (blue bars) and dry (red bars) conditions in the MLSHD according to the **(a)** SPEI1 and **(b)** dry episodes (orange bars) during 1980–2017.

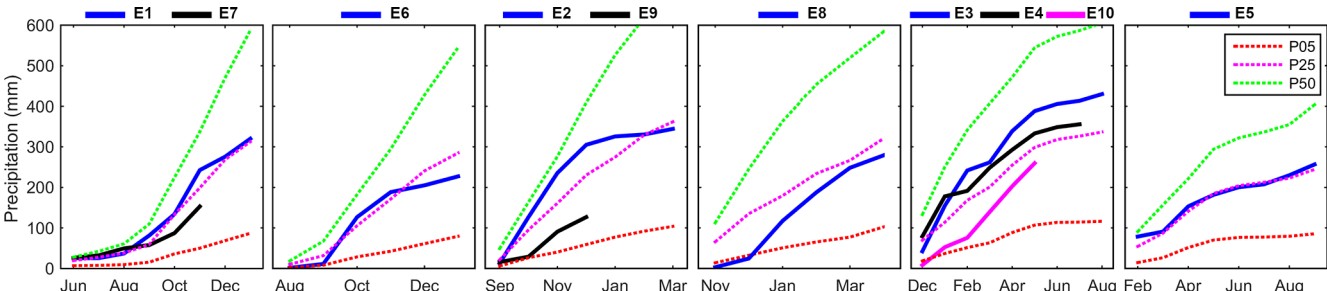

**Figure 6.** Accumulated precipitation during each drought episode listed in Table 3 (solid lines) and the monthly accumulated percentiles 5 (P05), 25 (P25), and 50 (P50) of precipitation (discontinued lines) while considering the whole study period (1980–2017).

**Table 3.** The 10 most severe drought episodes that affected the MLSHD from 1980 to 2017 arranged based on their severity from highest to lowest. The onset (first month of the episode), peak (lowest SPEI1 value during the episode), termination (last month of the episode), and duration (number of months from the onset to the termination) are shown.

| Episode | Onset | Peak | End | Peak value | Duration (months) | Severity |
|---------|-------|------|-----|-----------|-------------------|----------|
| E1 | Jun 2016 | Jul 2016 | Jan 2017 | −1.80 | 8 | 7.7 |
| E2 | Sep 2011 | Feb 2012 | Mar 2012 | −1.70 | 7 | 7.0 |
| E3 | Dec 2014 | Jun 2015 | Aug 2015 | −1.34 | 9 | 6.1 |
| E4 | Dec 2003 | Feb 2004 | Jul 2004 | −1.52 | 8 | 6.0 |
| E5 | Feb 1990 | Jul 1990 | Sep 1990 | −1.56 | 8 | 5.8 |
| E6 | Aug 1988 | Dec 1988 | Jan 1989 | −1.60 | 6 | 5.7 |
| E7 | Jun 2017 | Oct 2017 | Nov 2017 | −1.66 | 6 | 5.6 |
| E8 | Nov 2001 | Nov 2001 | Apr 2002 | −1.80 | 6 | 5.4 |
| E9 | Sep 2007 | Oct 2007 | Dec 2007 | −1.74 | 4 | 5.1 |
| E10 | Dec 1991 | Dec 1991 | May 1992 | −1.72 | 6 | 4.9 |

sis based on extreme meteorological dry conditions. In this table are also represented the onset, peak, end, peak value, duration, and severity of each episode. This information is also summarized for all dry episodes revealed by the SPEI1 from 1980 to 2017.

Through the SPEI it was not possible to know independently the role of the precipitation or an $Et_0$ value in the occurrence and magnitude of the drought. This is why in Fig. 6 the accumulated precipitation during each drought episode (solid lines, denoted as E1–E10) listed in Table 3 are illustrated, as well as the monthly accumulated percentiles 5 (P05), 25 (P25), and 50 (P50) of precipitation for the same climatological period (1980–2017; discontinued lines). The order of the episodes in this figure is determined by the month of their beginning. Accumulated precipitation during June and July of E1 was between the P05 and P25 but later was between P25 and P50. In E7 the accumulated precipitation was between P25 and P50 from June to August, and afterward, from September to November it was drier still (between the P05 and P25). For all 10 episodes, the accumulated precipitation was never above P50, confirming the precipitation deficit. However, in some episodes that are more severe than others, the accumulated value of precipitation is higher (e.g. E3, E4, and E10), indicating that potential evap-

otranspiration played a crucial role in determining the magnitude of drought.

In Fig. 7 six annual leading modes of the EOF are shown, which explained 94 % of the total spatial variability of the SPEI1 for those months when the SPEI1 $\leq -0.84$ in the MLSHD (represented in Fig. 5b). These represented potential physical modes of drought variability in the MLSHD. The first eigenvector (EOF1) explained 61 %. This pattern was very homogeneous, with close negative values in all the MLSHD. This indicates a great spatial homogeneity of the main drought pattern and the predominant influence of large-scale factors. Although, a visual analysis of the EOF1 also shows a small longitudinal difference with more intense negative values on the eastern part of the MLSHD (farther from the coast). As expected, the characteristics of the following EOFs represented in Fig. 7 show major spatial differences. The EOF2 explained 17 % of the total drought variability. In it, the major differences were observed between the eastern part of the region, where positive values prevailed, and the rest of the territory with negative values, indicating different spatial drought magnitudes. In Fig. 1 it can be observed that the eastern of the MLSHD is characterized by major elevation. Therefore, spatial drought variability in this pattern can be explained by orographic differences. The EOF3 ex-

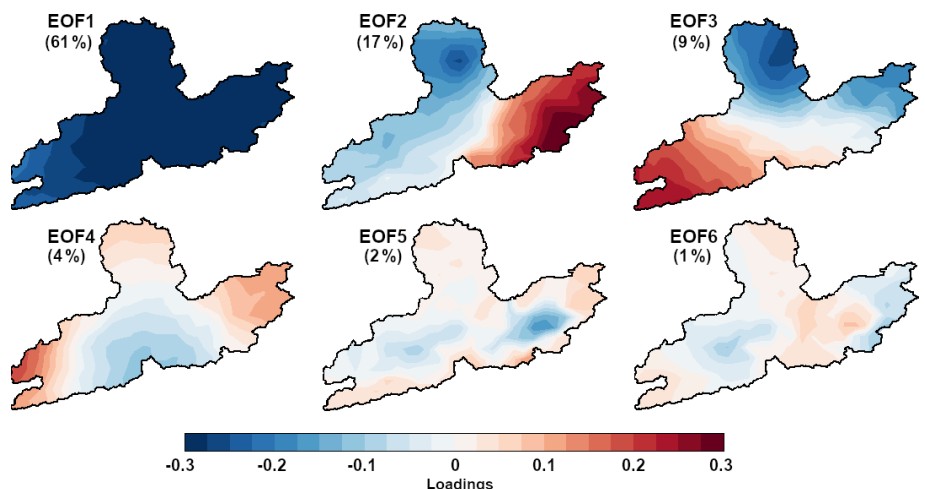

**Figure 7.** Six leading modes of the EOF for the months characterized by average SPEI1 values $\leq -0.84$. The numbers between parentheses correspond to the amount of variance explained by each EOF.

plained 9 % of spatial drought characteristics, which were determined by a gradient from positive to negative values from the coastal zone to the northeast, respectively. The remaining EOFs showed greater spatial variability and lower percentages of explained variance.

### 3.2 Relationship between the circulation weather type classification and drought conditions

The MSLP fields and anomalies for the 10 pure CWTs responsible for the major variability in atmospheric circulation over the IP are shown in Fig. 8. These anomaly composites were obtained after removing the respective means computed for the period of 1981–2010. Here we aimed to determine the association of large-scale atmospheric circulation over the IP with drought conditions that affected the MLSHD during 1980–2017. The reddish (blueish) isolines in Fig. 8 identify the higher (lower) values in the MSLP absolute fields and the positive (negative) values of MSLP anomalies. The NE configuration (Fig. 8a) was characterized by a transition from a strong high-pressure region over the eastern Atlantic Ocean extending to northwestern Iberia and the MLSHD, with lower pressures over Africa. The anomaly field (Fig. 8b) shows that this high-pressure centre was displaced towards the northeast, to the west of the United Kingdom (UK). Figure 8a shows that in the E configuration the high-pressure system was shifted northwards and centred over the Cantabrian Sea and the Celtic Sea, while in the SE circulation it was centred over France and the southern UK. The anomaly fields (E and SE; Fig. 8b) show an intensification of 8 to 10 hPa of these high-pressure systems. In the S pattern, higher pressure values occurred over central Europe, and lower pressure values (1010 hPa) occurred over the northeastern Atlantic (Fig. 8a), which were up to 8–10 hPa lower (Fig. 8b). In the SW CWT high-pressure values

were limited to the most southern areas in the North Atlantic and a well-developed low-pressure system (1000 hPa) was located over the northeastern Atlantic (Fig. 8a). The anomaly fields show an intensification of these systems to the northwestern region of Iberia – up to −20 hPa (Fig. 8b). In the W and NW configurations, the low-pressure systems were shifted northwards and northeastwards towards the UK, respectively, while the Azores High was established (Fig. 8a). The corresponding anomaly fields illustrate the intensification of these low-pressure systems (Fig. 8b). The high-pressure systems identified in the case of the NW configuration were more intense in the N CWT (Fig. 8a), and the anomaly shows a northward displacement of these systems, covering all the Atlantic regions, while low-pressure systems were more developed over the Gulf of Lion, in the Mediterranean (Fig. 8b). Finally, the C CWT represented low relative pressures located over the western IP (Fig. 8a) which intensified, while positive anomalies developed for the northern regions, west of the UK (Fig. 8b). The opposite occurred for the A configuration, which represents an intense Azores High, as it extended towards Europe (Fig. 8a). This anomaly shows that under these conditions the high-pressure systems intensified over the IP and southwestern Europe (Fig. 8b).

The correlations between the monthly percentage values of the occurrence of each of the pure CWTs with the SPEI1 time series are shown in Fig. 9. CWTs and SPEI1 time series were de-trended before correlation computation. The significant positive correlations found with SW, W, and NW CWTs are in agreement with the results of Russo et al. (2015), but they also relate to the entire NWIP. Indeed, air masses from the SW, W, NW, and C configurations are usually associated with inbound baroclinic structures, Atlantic storms, and atmospheric rivers (Eiras-Barca et al. 2018) which carry moisture from the Bay of Biscay (BB) and the tropical–subtropical North Atlantic corridor to the MLSHD, both of

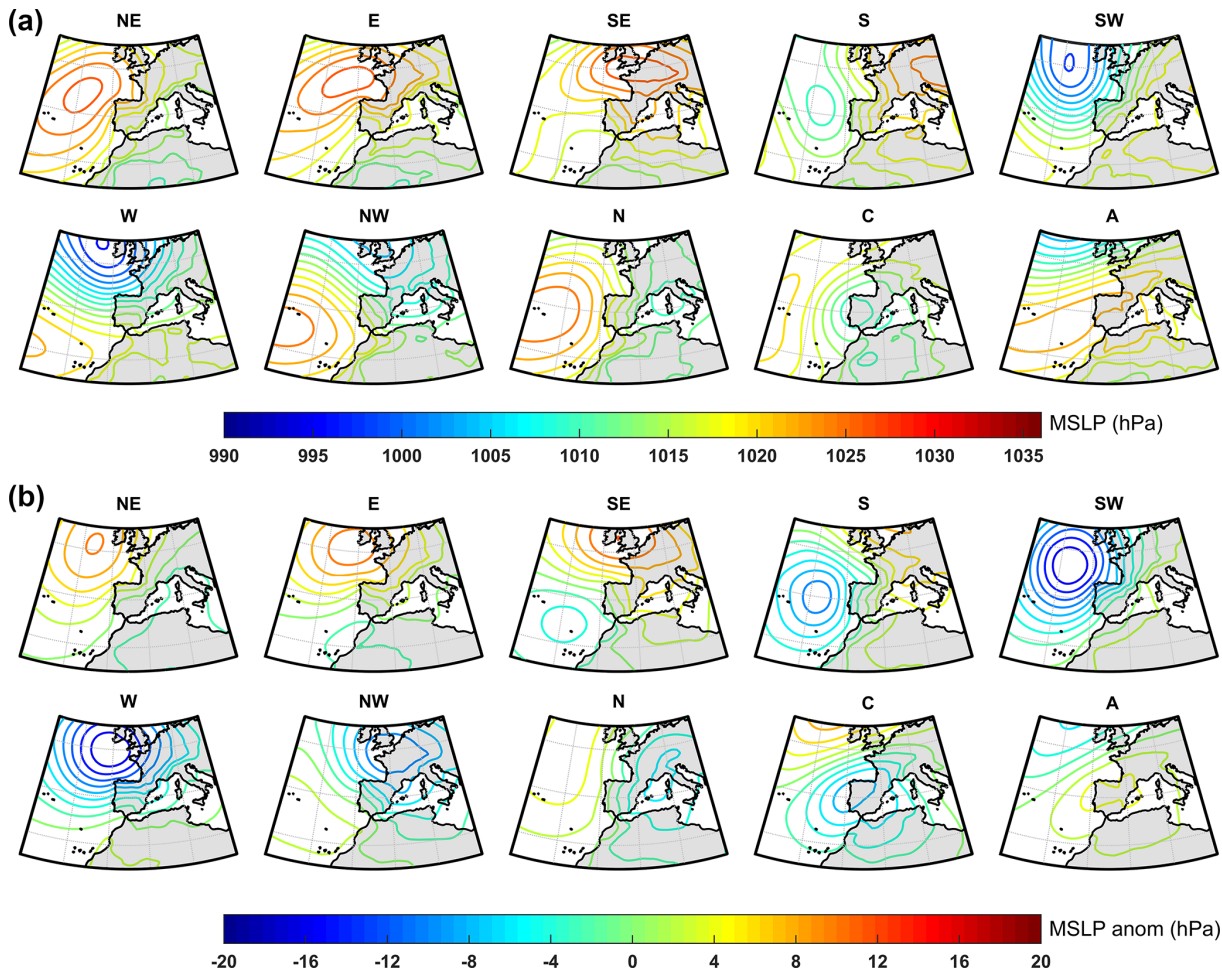

**Figure 8. (a)** Mean sea level pressure (MSLP) fields and **(b)** anomaly (anom) field configuration of the 10 pure circulation weather types (CWTs) for the period of 1980–2017. The contour interval is 2 hPa.

which are principal sources for precipitation over Galicia and northern Portugal (Drumond et al., 2011). The CWT appears to be mostly positively correlated with the SPEI1; however, for almost all months the correlations were not sta-
5 tistically significant. The extratropical cyclones and the associated synoptic-scale fronts reaching the IP during winter months and early spring normally produce large accumulated rainfall and are thought to play an important role in the hydrological cycle in northern Portugal and Galicia (Paredes et
10 al., 2006; Añel et al., 2014 TS29; Hénin et al., 2019).

Contrastingly, the atmospheric circulation associated with NE, E, and SE CWTs was negatively correlated with the SPEI1 time series in all months, thereby suggesting that air masses associated with these were directly related to the
15 dry conditions in the MLSHD. Negative correlations between the SPEI1 and the A CWT mostly occurred during winter months; however, these were lower and not significant during several of those months. On the contrary, the correlations between the SPEI1 and C configuration are mostly pos-
20 itive but mostly not statistically significant. Finally, as ex-

pected, monthly correlations between the atmospheric circulation associated with the N and S CWTs with the SPEI1 generally had opposite sign values, in addition to being very low and not statistically significant. Trigo et al. (2004) asso-
ciated the mean annual and seasonal rainfall decrease across 25
the IP during the second half of the 20th century to the lower occurrence of the high-rainfall circulation types (cyclonic) and with the increase of the low-rainfall types (anticyclonic). However, trend analysis for the period of 1980–2017 (not shown) revealed no statistically significant trend in the series 30
of any CWT.

The spatial patterns of correlations between the detrended series of the SPEI1 and the climatic teleconnections indices appear in Fig. 10. In most of the patterns, no different signs of the correlation were observed, coinciding with the sign 35
of the correlation shown in Fig. 9. This confirms that there was a homogeneous influence of each CWT on the variability of dry and wet conditions in the MLSHD according to the SPEI1. However, local variations of the correlation were still observed. For example, the spatial correlations of the SPEI1 40

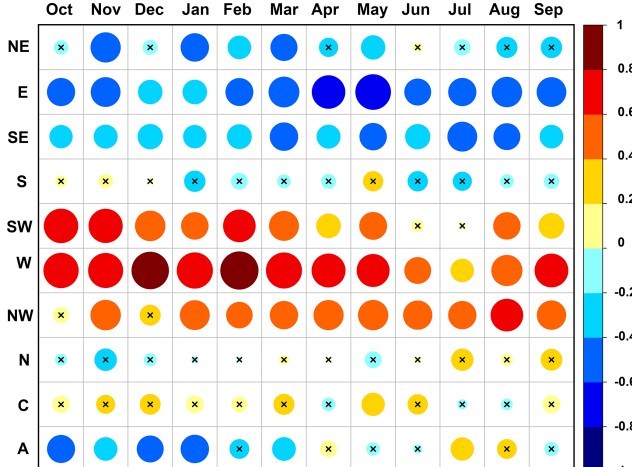

**Figure 9.** Correlations between the detrended series of the SPEI1 and the monthly percentage of occurrence of each of the pure CWTs for 1980–2017. The size of the circles is proportional to the correlation values. The x symbols inside the circles represent statistically non-significant correlations at $p < 0.05$.

with the W and E CWTs show a west–east gradient from highest to lowest values. The correlations were statistically significant throughout the MLSHD only for the SE, E, SW, W, and NW CWTs.

In order to understand how distinct CWTs might have affected drought severity in the MLSHD, Fig. 11 shows the monthly frequency (expressed in percentage) of each CWT under different drought categories (moderately dry, severely dry, and extremely dry) according to the SPEI classification shown in Table 1. Those Octobers under moderately dry conditions were associated with the prevalence of A, E, and C CWTs. Octobers affected by severely dry conditions were associated with a major percentage of A circulation, but for those under extreme drought conditions, it seems that E circulation highly increased with respect to previous drought categories, while there was a slight decrease in the frequency of A circulation. For those Novembers affected by moderate, severe, and extreme drought conditions, the most frequent CWT was the A circulation, which imposed an atmospheric flux from the north. For severely and extremely dry Decembers the frequency of CWTs changes with respect to those of previous months, and an increase in the percentage of SE circulation was observed. Januaries under moderate and severe drought conditions were characterized by a major percentage of atmospheric conditions governed by the A pattern. In February, the percentage of occurrence of the A CWT decreased when drought severity increased, while the E CWT increased for severely drought months, and the NE CWT increased for extreme drought months. For those Marches under drought conditions, the most frequent patterns were the A, E, and NE CWTs. The frequency of CWTs for those Aprils affected by drought conditions was remarkably dif-

ferent with respect to those described for previous months. In these months the E and SE CWTs were directly related to drought severity increase. The following months (May to September) were affected by different drought categories; the combination of NE, E, and A CWTs was the most frequent according to the percentage observed in Fig. 11.

Figure 12 shows the accumulated SPEI1 (red line) during the 10 most severe drought episodes listed in Table 3. The coloured areas in this figure represent the CWTs that occurred for every day of the episode. CWTs were grouped taking into account the monthly correlation results presented in Fig. 9. A visual analysis found that, along with the temporal evolution of all episodes, the most frequent CWTs were the eastern (NE–E–SE; yellow colour) and A (orange colour) CWTs. In agreement with results thus far described, for most of the episodes, the eastern circulation seems to be especially related to the drought intensification, being the most common CWT during the peak month of each event. Western circulation patterns appeared randomly during the episodes. In the last days of E1, E5, and E6, the SW, W, and NW CWTs were observed, while in the last days of E2, E3, and E7, the C CWT was observed.

The anomaly in the percentage of occurrence of every pure CWT during the 10 most severe episodes is shown in Fig. 13. The anomaly was calculated for the complete duration of each drought episode and referred to the 1980–2017 mean value for the same months. The eastern (NE, E, and SE), A, and S CWTs experienced mostly positive anomalies, in accordance with the results described for Fig. 12. Conversely, in most of the episodes the anomaly of the western-circulation (NW, W, SW), N, and C CWTs decreased. The largest negatives anomalies appeared for the W CWT, which had an average between 2.5 % and 5.7 %. Similar results were observed when the total number of severe events was considered (Fig. S1 in the Supplement).

Figure 14 shows the anomaly of the VIMF and its divergence for the onset, peak, and termination of drought episodes listed in Table 3. E1 was the driest and was associated with anticyclonic circulation of the VIMF with its centre located to the southwest of the MLSHD. This centre moved to the north during the peak of the episode, imposing moisture flux anomalies from the northeast. This was supported by prevailing A and NE CWTs, which decreased in percentage when the drought disappeared (February 2017), in accordance with the major frequency of C, W, and SW circulation and negative anomalies of the VIMF divergence (favouring the convergence). E2 began in September 2012. For this month, intense positive anomalies of the VIMF divergence over the MLSHD were observed, which were a dynamic limitation for the occurrence of precipitation. The A and NE CWTs were the most frequent CWTs during that month. The peak of this episode occurred in February 2012, when intense anticyclonic anomalies of the VIMF with its centre near the southwest of Ireland dominated the North Atlantic sector. In accordance, NE and E circulations were the most frequent

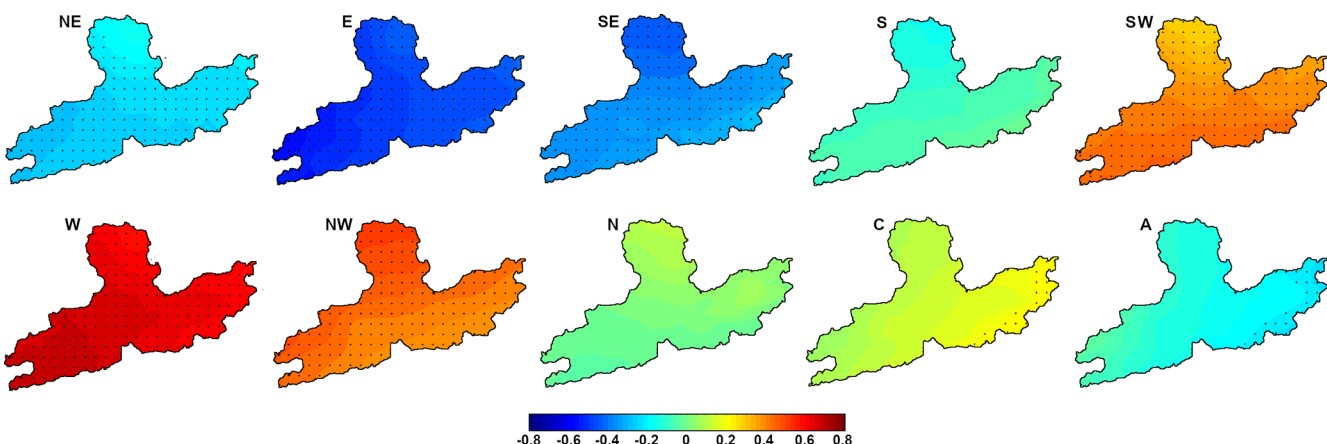

**Figure 10.** Spatial correlations between detrended grid series of the SPEI1 at a resolution of 0.1° in longitude and latitude and detrended monthly percentages of occurrence of each of the pure CWTs for 1980–2017. The dots in every correlation map denote statistically significant correlations at $p < 0.05$.

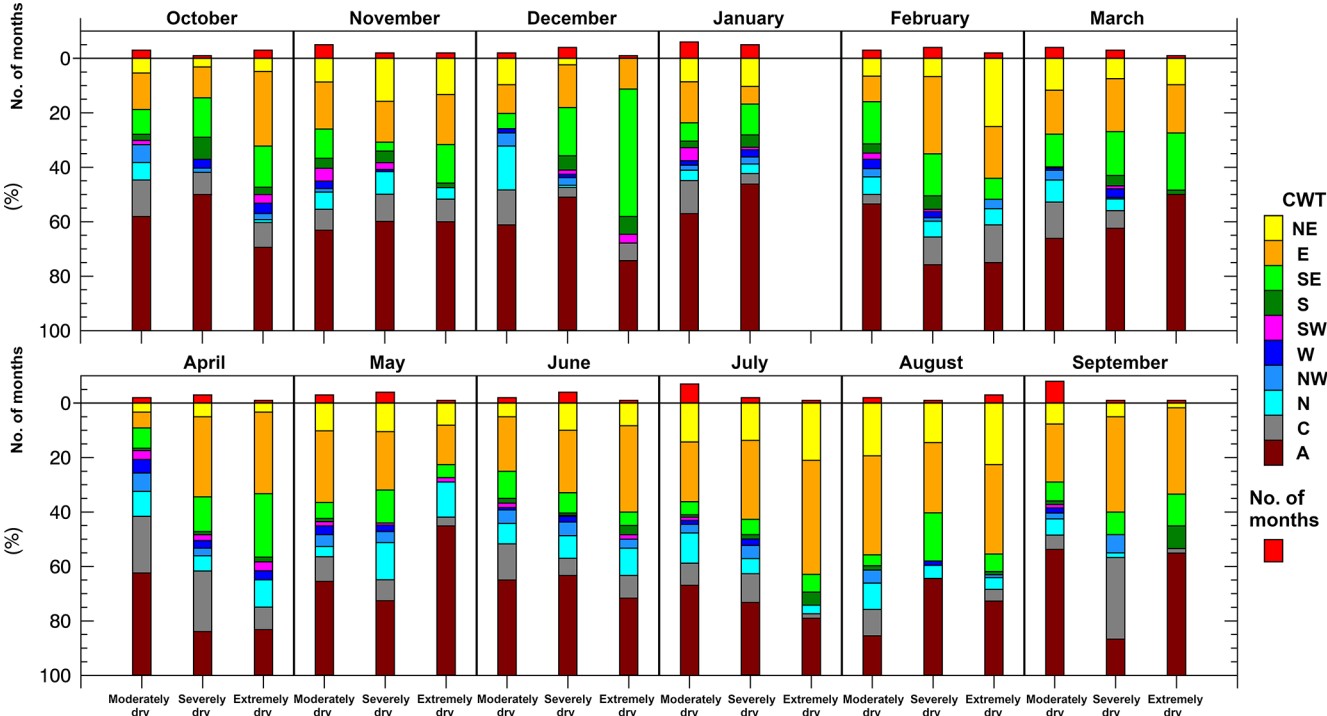

**Figure 11.** Monthly percentage of occurrence for every CWT associated with moderately, severely, and extremely dry conditions. The red bars represent the number of months the MLSHD was affected by each drought category.

over the IP. Drought conditions disappeared (in April 2012) when negative anomalies of the VIMF divergence in association with cyclonic circulation anomalies of the VIMF with its centre located over England affected the MLSHD. Correspondingly, the most frequent circulation patterns during that month were the C and NW CWTs. The third and fourth driest episodes began in December of 2014 and 2003, respectively. In both months, the A CWT was the most prominent circulation pattern. Anticyclonic anomalies of the VIMF with its

centre over the North Atlantic affected the MLSHD; these anomalies were more intense in December 2014, when positive anomalies of the VIMF divergence covered almost all of the IP. The peak of E3 occurred in June 2015 with intense VIMF anomalies from the Atlantic Ocean that reached the northern portion of the IP; however, over the MLSHD, both negative and positive VIMF divergence anomalies were observed. The last month of E3 was August 2015 because the SPEI changed to a positive value in September 2015 ow-

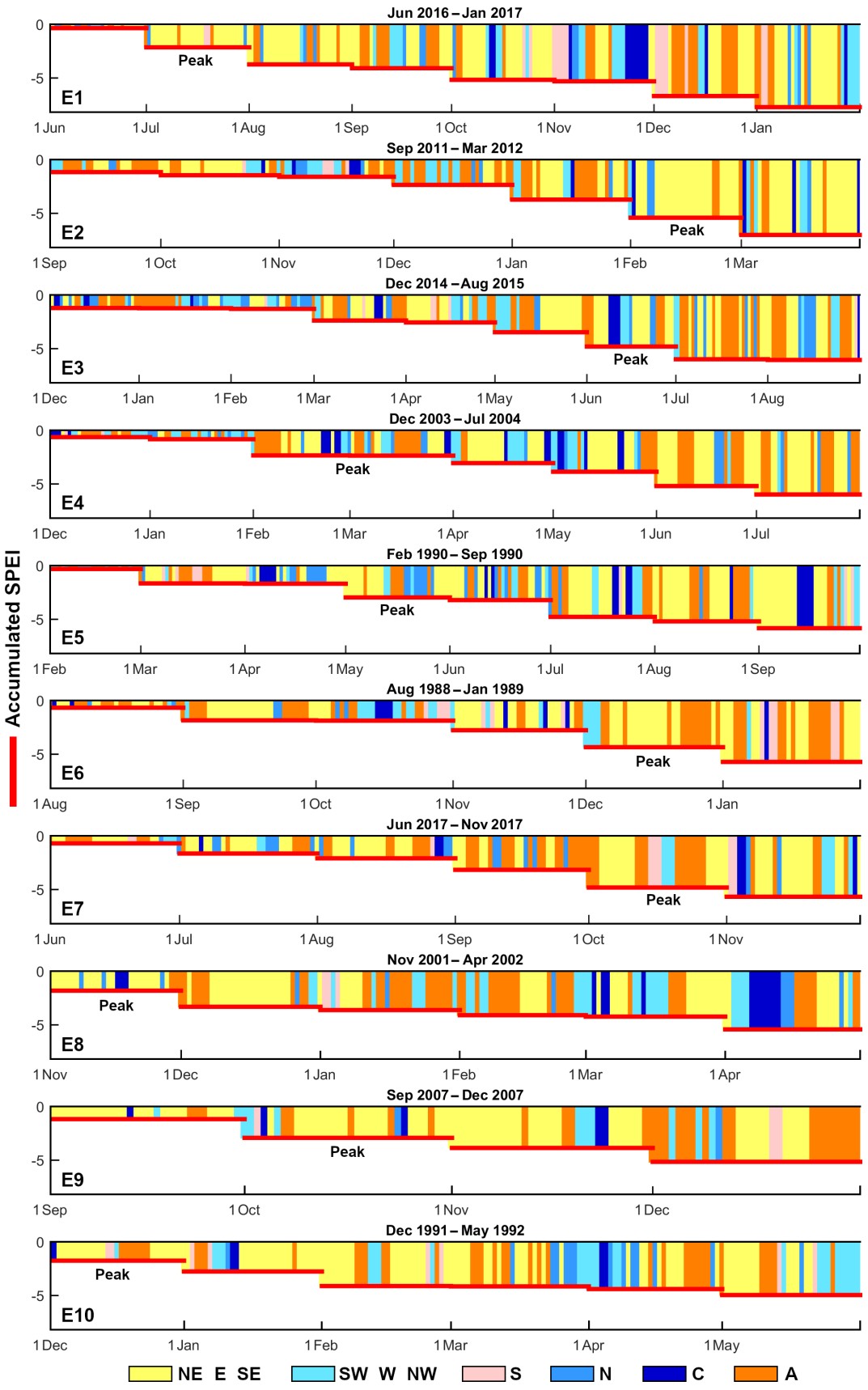

**Figure 12.** Accumulated SPEI1 (red line) and grouped CWTs during the 10 most severe drought episodes listed in Table 3.

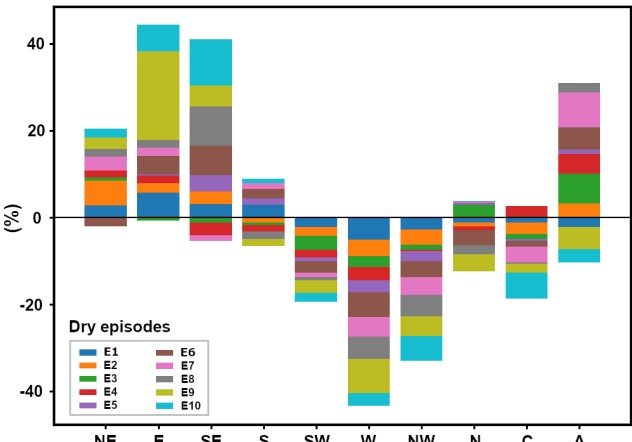

**Figure 13.** Anomalies in the percentage of each CWT associated with the 10 most severe drought episodes listed in Table 3.

ing to negative anomalies of the VIMF divergence over the NWIP and the influence of VIMF anomalies reaching the MLSHD from the northwest. This was in accordance with an increase in the percentage of the W CWT with respect to that in the previous stage of the episode. In the peak of E4, positive anomalies of the VIMF divergence over the ML-SHD were observed. This episode ended when the anomalies on the moisture flux from the west favoured the occurrence of convergence, despite the fact that the most frequent CWT was the A CWT, followed by the W CWT. E5 began in February 1990. In that month, the VIMF anomalies over the IP showed an intense anticyclonic circulation accompanied by positive divergence anomalies over all the IP and consequently, a high frequency of A circulation. In the peak there is no clear pattern of VIMF divergences; however, 1 month after the demise the SPEI1 became positive due to negative anomalies of the VIMF divergence and enhanced moisture flux reaching the MLSHD from the northwest.

In August 1988 the VIMF anomalies in the onset of E6 were characterized by an anticyclonic circulation centre to the southwest of the MLSHD and cyclonic circulation in the northwest, which were both over the Atlantic Ocean (Fig. 14) and enforced VIMF anomalies reaching the ML-SHD from the west. Nevertheless, the VIMF divergence anomalies showed prevailing divergence conditions. That is, the MLSHD could receive air masses from the west, which, as already described, were associated with the increase of wet conditions, but dynamic atmospheric conditions could inhibit the occurrence of precipitation. In the peak of this episode (December 1988), a centre of anticyclonic circulation of the VIMF was observed in the North Atlantic Ocean to the northwest of the IP that produced an intense divergence of the VIMF over the MLSHD. The predominant frequency of the A and E circulations were also observed for that month. That episode ended when VIMF anomalies reached the MLSHD from the northwest, in combination with negative anomalies

of the VIMF divergence over the MLSHD. In the onset of E7, as well as in E6, a centre of anomalous cyclonic circulation of the VIMF was observed in the North Atlantic Ocean to the northwest of Ireland. That situation was different at the peak of the episode (October 2007). In that month, anticyclonic circulation of the VIMF with the centre located to the northwest and near the MLSHD imposed strong divergence of the VIMF over the MLSHD, while the prevailing frequency of A and E CWTs occurred. In December 2017 this centre was located further west, and the VIMF divergence anomalies became negative over the MLSHD, while the SPEI1 turned out to be positive. In E8, the onset (November 2001) coincided with the peak of the episode. That month was characterized by the prevalence of the E and NE CWTs and positive anomalies of the VIMF divergence associated with the strong anomalous anticyclonic circulation of the VIMF over the North Atlantic Ocean. That episode ended in April 2002 owing to a positive SPEI1 1 month after the demise (May 2002). A small area of negative VIMF anomalies over the MLSHD was observed for that month. As well as in the previous episodes, the onsets of E9 and E10 were characterized by anticyclonic anomalies of the VIMF with the centre located to the northwest of the MLSHD over the Atlantic Ocean. The strong VIMF divergence was observed over the MLSHD in the peak of E9 and the onset of E10. The onset and peak of E10 coincided. The VIMF anomaly pattern for that month (December 1991) was characterized by anticyclonic circulation with the centre located in the North Atlantic Ocean to the northwest of the MLSHD near Ireland. This pattern was very similar for the peaks of E9, E2, E5, E6, and E7. Both episodes E9 and E10 ended owing to negative anomalies of the VIMF divergence (especially for E10).

### 3.3 Relationship between drought and modes of climate variability

Figure 15a shows the correlation between the BEST, NAO, EA, AO, SCAND, and WeMOi series with the SPEI1 up to the SPEI24. This analysis helps to determine any causal effect between atmospheric and oceanic teleconnection patterns and dry and wet conditions in the MLSHD. The results reveal a major link between the SCAND (positive correlation) and AO (negative correlation), particularly at short temporal scales of the SPEI (1 to 4 months). The SCAND pattern (initially referred to as the Eurasia-1 by Barnston and Livezey, 1987) in its positive phase was characterized by positive height anomalies over Scandinavia and western Russia but weaker centres of the opposite sign over southern and western Europe. The negative phase showed the opposite pattern. The results of Rodriguez-Puebla et al. (1998) show heterogeneous spatial correlation between the SCAND index and the annual precipitation over the IP; however, they confirmed that annual precipitation variability in the northwestern IP was related to the SCAND December pattern. On the contrary, another study showed that the AO ranged from

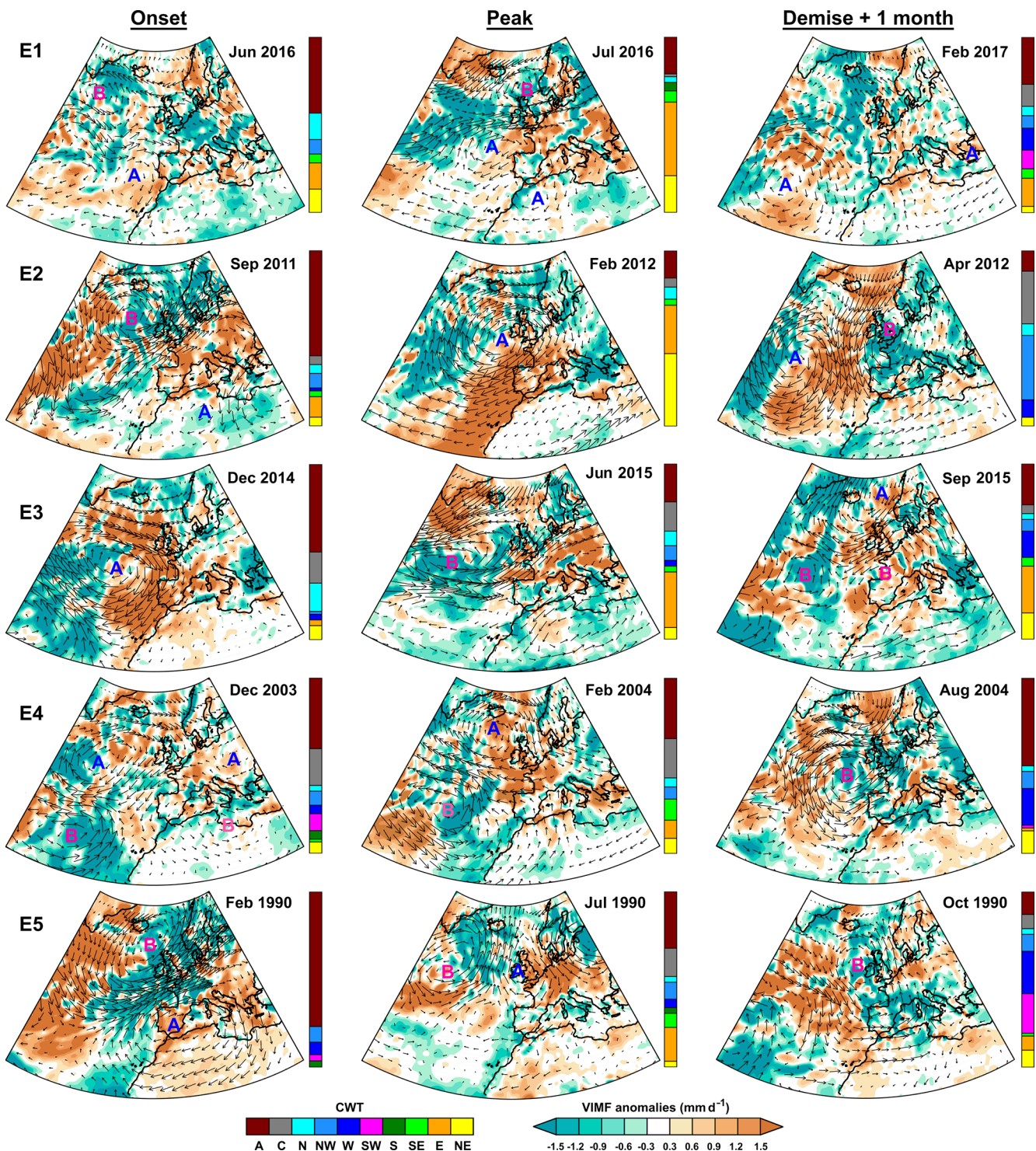

**Figure 14.**

positive to negative values depending on pressure anomalies in the Arctic region (Thompson and Wallace, 1998). A band of strong winds circulating around the North Pole associated with the positive phase of the AO kept colder air within the polar region and corresponded to a deepening of the Azores High and the strengthening of the polar and subtropical jets over the Euro-Atlantic region (Ambaum et al., 2001). In the negative phase, this ring is thought to become weaker, thereby allowing the southwards penetration of Arctic air masses and an increase in the magnitude of the total

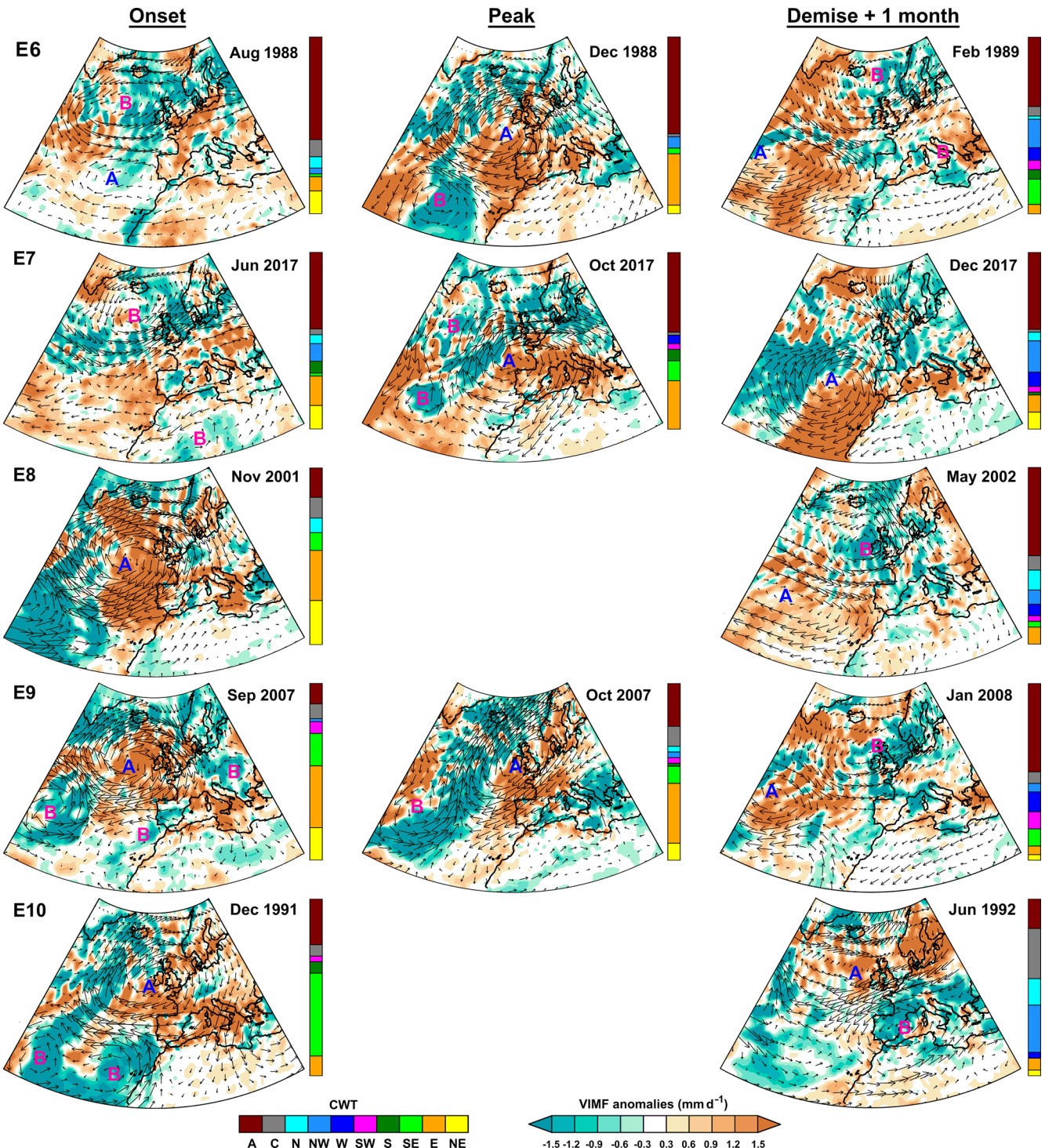

**Figure 14.** The monthly anomaly of the VIMF (in arrows) and its divergence (shaded) during 1980–2017 for the onset, peak, and 1 month after the termination of each of the 10 most severe drought episodes (Es) listed in Table 3. Anticyclonic (or cyclonic) centres of the VIMF anomalies circulations are represented as A (or B). Vertical bars show the monthly percentage of each pure circulation weather type (CWT).

eddy energy fluxes into the Euro-Atlantic region (Rivière and Drouard, 2015), which would affect the hydroclimatic conditions in the northwestern IP (deCastro et al., 2006a) and this explains the negative correlations obtained with the SPEI in this study.

According to Wanner et al. (2001), the AO is similar to the NAO in many aspects. Multiple results have shown a strong relationship between the winter tropospheric pattern of the NAO and AO (Wanner et al., 2001; Rogers and McHugh, 2002; Hurrell et al., 2003 **TS30**; Dai and Tan, 2017). However, although the AO is strongly correlated with the NAO, it does not show the recent sustained significant summer decrease, but it does show enhanced early-winter variability (Hanna et al., 2015). The results of Tabari and Willems (2018) show that the AO signal is oppositely related to anomalies of daily precipitation extremes during summer in the NWIP, a phenomenon they did not see occurring with the NAO. Therefore, in order to assess any possible difference in the impacts over drought conditions, both indexes were used in this investigation. The negative phase of the NAO was associated with the weakness of the Azores High and a southwards position in the storm tracks, thereby resulting in wet conditions over the IP (Trigo et al., 2002). The correlations in Fig. 15a demonstrate that both the NAO and AO relationships with the SPEI were nearly identical across different temporal scales. However, the correlations with the AO are greater, indicating that the AO index may be more effective for explaining the atmospheric influence on dry and wet conditions in the MLSHD. Nevertheless, the NAO index has also been traditionally defined as the normalized pressure difference between a station on the Azores and one on Iceland (Hurrell, 1995; Jones et al., 1997); therefore, the correlations with the SPEI could be also different in this regard.

The correlations between the SPEI and the BEST index are positive but also very low ($< 0.2$) and not significant; moreover, these became negative when correlations were made with SPEI values computed from the past 6 to 24 months but were also not statistically significant (Fig. 15a). ENSO is namely the strongest ocean–atmosphere coupling phenomenon on the interannual timescale, but our results suggest a poor association between ENSO (El Niño and La Niña) and the occurrence of dry and wet conditions in the MLSHD. The findings of García et al. (2005) revealed that ENSO influence was not significant on the precipitation over Galicia. Though, according to Dai and Tan (2017), a warm (cold) ENSO enhanced the negative (positive) AO phase, which is directly related to the MLSHD hydroclimate. Finally, the correlations between the SPEI1 to the SPEI24 with the WeMOi and EA were positive but only statistically significant for the WeMOi within the first two temporal scales of the SPEI.

Because the correlations in Fig. 15a are greater with the SPEI1 than with other temporal scales of this index, a second correlation analysis was conducted in order to determine the relationships between the SPEI1 and the teleconnections phenomena but at monthly scale (Fig. 15b). The correla-

tions with the BEST index are mostly not statistically significant. Negative correlations only occur in spring (March, April, and May; Fig. 15b). To the contrary, Muñoz-Díaz and Rodrigo (2004) found that the negative phase of ENSO, La Niña, leads to a low probability of drought in spring but for the whole northern IP, while Lorenzo et al. (2010) also concluded that La Niña almost always announces dry springs in the NWIP. Unlike these authors in this study, we use an index that contemplates ocean and atmospheric conditions to identify the phases of ENSO, and another index that contemplates both precipitation and evapotranspiration to identify drought conditions. In any case, this issue deserves further study.

The monthly correlations obtained between the NAO and AO with the SPEI1 were very similar (Fig. 15b); however, as expected from the results already described for Fig. 15a, the AO was best related with the SPEI1 series and consequently to monthly dry and wet conditions in the MLSHD throughout the hydrological year, especially in the winter and spring months (December to March). This is in agreement with Manzano et al. (2019), who argued that the AO and NAO patterns have a significant impact on droughts in winter over large areas of the IP. However, at local and regional scales, results may differ. In a previous study by Rodríguez-Puebla and Nieto (2010), it was revealed that positive (negative) NAO induced an east–west decreasing gradient of drier (wetter) conditions over the IP. The most recent findings of Sáez de Cámara et al. (2015) describe a complete lack of correlation between $P$ anomalies and the NAO for the central and eastern parts of the northern IP. These authors also have shown that from the late 1980s to 2005 an increase occurred in the frequency of extreme circulation modes within each of the NAO positive and negative phases, both inducing negative precipitation anomalies and a long-lasting dry spell in the northern IP. That is, special consideration must be made when associating a positive trend of the NAO with the increase of dry conditions over the entire IP. In this study, positive correlations between the EA and SPEI1 were observed in the winter and spring months (November to May). Results of Casanueva et al. (2014) also revealed a positive correlation of the EA with the $P$ values and the consecutive wet days over the NWIP during the boreal winter. The SCAND pattern was also positively correlated during all months of the year, but no significant correlations were found in December, February, and March. This means that the negative phase of the SCAND pattern was related to dry conditions in the MLSHD. During the negative phase, the European trough was thought to deepen, while weak pressure ridges were observed over the northeastern Atlantic Ocean (Bueh and Nakamura, 2007). This caused the Atlantic storm track to extend northeastwards, affecting a vast area from northern Europe to central Siberia. Finally, in this study the WeMOi showed positive significant correlations with the SPEI1 from January to November but especially during the boreal summer months. Previously, the WeMOi had been associated with the precipi-

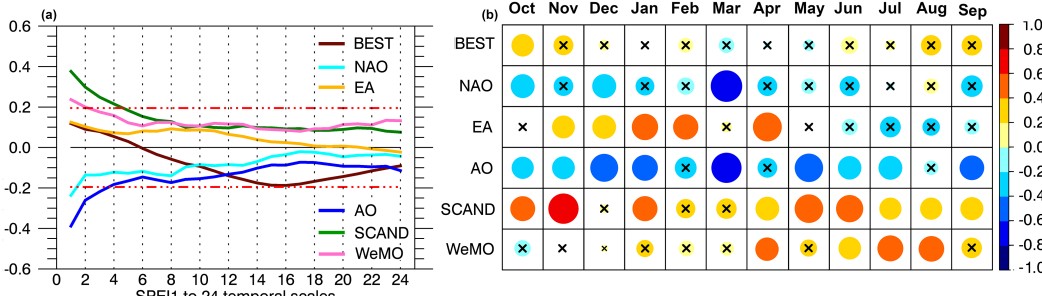

**Figure 15.** Correlation between the monthly series of the bivariate El Niño–Southern Oscillation (BEST), North Atlantic Oscillation (NAO), East Atlantic (EA), Arctic Oscillation (AO), Scandinavian pattern (SCAND), and the Western Mediterranean Oscillation (WeMO) with **(a)** monthly series of the SPEI1 to the SPEI24 and **(b)** the monthly correlation between the same climatic indices and the SPEI1 for the period of 1980–2017. Statistically non-significant correlations at the 95 % confidence level are those within discontinuous red lines in **(a)** and marked with an x in **(b)**. The size of each circle is proportional to the correlation value.

tation variability in the eastern part of the IP and the south of France (Martín-Vide and Lopez-Bustins, 2006; Martin-Vide et al., 2008). However, these correlations (Fig. 15b) indicate that this index could be also useful for explaining dry and wet conditions in the NWIP. In this study, the positive phase of the WeMO corresponded to the anticyclone over the Azores which may have transported moisture entering the MLSHD from the west, and as previously explained, the west circulation favoured the occurrence of wet conditions in the MLSHD, which is in agreement with the positive correlations found. In its negative phase, the WeMO is thought to coincide with the low pressure often cut off from northern latitudes, in the framework of the Iberian southwest (Martín-Vide and Lopez-Bustins, 2006). It possibly favours the east circulation over the NWIP, which may also explain the positive correlation between the WeMOi and SPEI1 noted in this study.

The frequency bands and time intervals of the co-variations between the SPEI1 and different modes of climate variability represented by climatic indices (i.e. BEST, EA, WeMOi, NAO, AO, and SCAND) are shown in Fig. 16. The coloured shading displays the magnitude in the coherence as represented in the colour bar, which varies from 0 to 1 and indicates the timescale variability in the correlation between the two time series. Warmer colours (red) represent regions with significant interrelation, while colder colours (blue) signify lower dependence between the series. The results reveal that the BEST index showed strong but intermittently significant interannual coherence with the SPEI1 in the period of the 1–7-month band. Moreover, a significant correlation was observed from 1980 to 1990 for the 40–64-month band, but it was outside the cone of influence (COI) until the end of 1983. In this timescale, the arrows pointing straight down indicate that the SPEI led the BEST index by 90°. In the case of the EA there was a frequent, significant co-oscillation with the SPEI1 in the high-frequency 0–6-month band. However, from approximately the end of the 2000s to 2012, a high co-

herence peak occurred in the low-energy regions (for nearly 30–50 months). The coherence between the SPEI1 and WeMOi exposed frequent but non-stationary interannual coherence regions at 1–8 months. At ∼ 66-month frequencies a strong positive coherence was noticed within the COI between 1992 and 2008.

The findings of Hurrell (1995) revealed that the NAO has a rich combination of high frequencies from intraseasonal to interannual timescales and a low frequency from decadal to multidecadal timescales. It had significant coherence with the SPEI at high frequencies (6 to 16 months) in the periods of 1982–1984, 1986, and 2004–2012 (Fig. 16), coinciding with dry periods in the MLSHD. Strong coherence was also noticed at a longer temporal scale (30 to 34 months) for the period of 1986–1993. Results of García et al. (2005) also suggest that the NAO and precipitation in Galicia could be related at a timescale of 8 years. For this study, compared with those in the NAO, oscillations in the AO were manifested in the SPEI1 over most of the period on intermittent wavelengths from 2 to 6 months but most significantly and for longer periods in the range of 6 to 36 months (3 years). In this frequency band, the left-pointing arrows show an anti-phase relationship (negative correlation), thereby indicating that the AO and SPEI1 moved in opposite directions from each other (i.e. when one was maximum, the other was minimum and vice versa). This is in accordance with previous correlations shown in Fig. 15. Finally, the significant coherence between the SPEI1 and the SCAND pattern reveals the influence of this teleconnection pattern between 1990 and 2005 along with the 0–8-month periodic bands and at low frequencies (from approximately 14 to 40 months) during longer and continuous periods. Alternatively, arrows pointing to the right and downwards and right and upwards indicate alternately the SPEI leads or lags and the SCAND pattern.

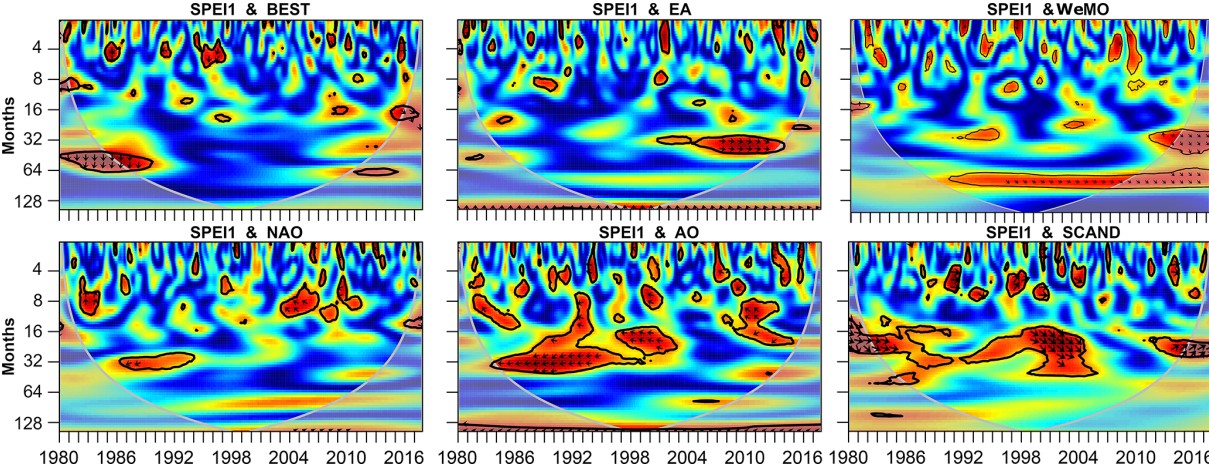

**Figure 16.** Wavelet coherence between the SPEI1 and the series of teleconnection patterns, namely the bivariate El Niño–Southern Oscillation time series (BEST), East Atlantic (EA), North Atlantic Oscillation (NAO), Arctic Oscillation (AO), and Scandinavian pattern (SCAND). The colours from blue to red indicate the increasing coherence. Areas enclosed by a black line correspond to statistically significant cross-wavelet powers at the 95 % level. The grey line depicts the cone of influence (COI), while the black arrows indicate the phase condition. The phase relationships between the climate indices and SPEI1 are denoted by arrows for the in-phase pointing right, anti-phase pointing left, climate indices leading the SPEI1 by 90° pointing up, and the SPEI1 leading the climate indices by 90° pointing down.

## 3.4 Hydrological drought

The temporal evolution of the SRI for temporal scales of 1 (SRI1) to 6 (SRI6) months appears in Fig. 17. Negative values of the SRI indicated runoff droughts, normally recognized as hydrological droughts. The high variability of the SRI1 makes it difficult to observe whether or not these occurred in continuous dry periods. The SRI6 better depicts the identification of continuous dry periods such as 1991–1993, 2004–2005, 2011–2012, and the end of 2006 to 2007.

In this section we present our investigation into the possible response of hydrological drought through the SRI1. This was decided considering the effect of current and previous drought conditions revealed by the SPEI1 to the SPEI24 series. Correlations values in Fig. 18a show that the SRI1 variability was well associated with the first temporal scales (1 and 2 months) of the SPEI along all of the hydrological year. However, high correlations during all SPEI temporal scales were observed for December, January, and February, thereby suggesting that surface runoff during the rainiest months may have also depended on dry and wet conditions from previous months. From April to September the highest correlations were more restricted to the previous 2–4 months. According to a statistically significant correlation in July (i.e. the climatological driest month), the surface runoff variability was also affected by dry and wet conditions from the previous 4–21 months. Moreover, the SPEI was based on a certain water balance; therefore, it would stand to reason that the runoff may vary directly with the associated $P$ annual cycle in the MLSHD. The maximum correlations in this figure indicated the best climatic timescale over which the runoff drought was measured by the SRI and responded

to dry and wet conditions according to the SPEI. However, drought propagation through every component of the hydrological cycle depends on the severity of drought as well as the characteristics of the catchments (Van Lanen, 2006).

Figure 18b illustrates the monthly response rate (in percentage) of hydrological drought (SRI $\leq -0.84$) to drought at different timescales according to the SPEI1 to the SPEI24 being less than or equal to $-0.84$. Dry conditions revealed at all temporal scales of the SPEI had remarkably different response rates of runoff drought across the hydrological year. The larger responses ($> 50$ %) occurred from October to April and particularly in January, February, and March (i.e. the rainiest months). For these months, the large rate of months under hydrological drought was highly affected by drought conditions from several months before. This is in agreement with the correlation shown in Fig. 18a. From May to September the $P$ values over the MLSHD decrease, and the rate response of runoff drought reaches $\sim 20$ %. In these months the runoff droughts are best linked to drought conditions at shorter SPEI temporal scales; as precipitation declines, the $Et_0$ value is a determinant factor in the modulation of soil moisture content and runoff.

## 4 Conclusions

In this study drought phenomena in the MLSHD were investigated through the SPEI using high-resolution gridded datasets. An application of the EOF method revealed highly homogeneous drought features despite the complex topography of the region. During the period of the study (1980–2017), frequent meteorological droughts af-

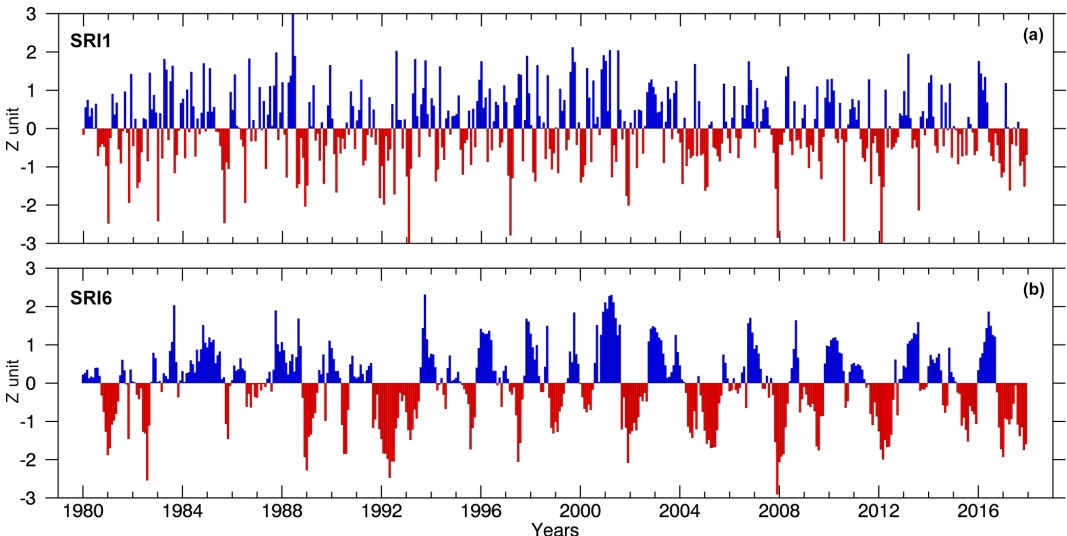

**Figure 17.** Temporal evolution of the standardized runoff index computed for **(a)** 1-month and **(b)** 6-month temporal scales in the MLSHD. Period of 1980–2017.

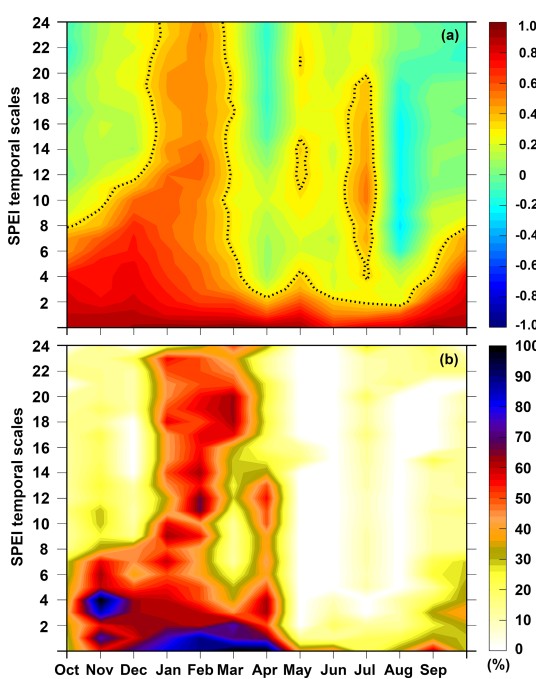

**Figure 18. (a)** Monthly correlations among the SRI1 for the entire Miño–Limia–Sil hydrographic demarcation (MLSHD) with the SPEI1 to the SPEI24 in the MLSHD; values within dotted lines represent significant correlations at $p < 0.05$. **(b)** Rate (in percentage) of hydrological drought (SRI1 $\leq -0.84$) during drought conditions according to the monthly SPEI1 $\leq -0.84$. Period of 1980–2017.

fected the MLSHD in 1989–1992, 2004–2007, and 2015–2017, while the most severe drought episodes occurred during June 2016–January 2017, September 2011–March 2012, December 2014–August 2015, etc. (Table 3). To investigate the atmospheric circulation associated with different drought categories in the MLSHD, a CWT classification for the entire IP was used. The results confirm previous findings for the northwestern and entire IP, and they showed that the MLSHD is a hydroclimatic region where atmospheric circulation associated with weather types SW, W, and NW (NE, E, and SE) is related to wet (dry) conditions. A spatial correlation analysis between 10 pure CWTs and the SPEI series computed on a 1-month temporal scale revealed a highly uniform influence of every CWT (and therefore the associated circulation) on the spatial variability of dry and wet conditions.

We found that dry and wet conditions in the MLSHD were susceptible to external forcing not only in the short term but also for the mid-to-long-term changes. The most influential teleconnection patterns on the variability of dry and wet conditions in the MLSHD were the AO and SCAND pattern, followed by the NAO, which is in agreement with previous results for the region. The signals of the AO and NAO were opposite of the SPEI1 in the MLSHD, while contrastingly, the SCAND pattern was positively correlated with the SPEI1 series. Several studies have recognized the NAO as the dominant pattern for the Euro-Atlantic region. A more detailed study on the short-, medium-, and long-term impacts of the NAO and AO on the atmospheric dynamics associated with hydrometeorological extremes in the NWIP should be conducted. Similar to the SCAND pattern, the WeMOi was also positively correlated with the SPEI1 in the MLSHD. Intermittently significant coherence between the SPEI1 and other teleconnection patterns (i.e. BEST and EA) was also detected

in the high-frequency region, but statistically significant correlations indicated there was not a strong relationship of the ENSO event and the EA mode on the water balance in the MLSHD.

The SRI was used as a complement to the SPEI for representing hydrological drought in the MLSHD. We found that a fast propagation of meteorological drought to runoff drought exists across the year; normally at very short timescales (1–2 months). However, this influence was higher in the climatological rainiest months of the year (winter months), when hydrological drought was affected by the previous 24 months of drought according to SPEI values less than or equal to $-0.84$. This relationship was less observed in the dry season. In conclusion, this study provides information that is fundamental to understanding the climate forcing of dry conditions in the MLSHD, which is an important hydrological and socioeconomic region of the NWIP. Furthermore, these results will support hydrometeorological forecasting and water management plans for the region.

*Data availability.* . TS31

*Supplement.* The supplement related to this article is available online at: https://doi.org/10.5194/nhess-20-1-2020-supplement.

*Author contributions.* . TS32

*Competing interests.* The authors declare that they have no conflict of interest. TS33

*Special issue statement.* This article is part of the special issue "Hydroclimatic extremes and impacts at catchment to regional scales". It is not associated with a conference. TS34

*Acknowledgements.* Rogert Sorí acknowledges support from the European Union and FEDER through the RISC_ML CE11 project under the INTERREG España–Portugal programme. Rogert Sorí and Milica Stojanovic acknowledge the grants received from the European Association for Territorial Cooperation, Galicia Northern Portugal (AECP, GNP) through the IACOBUS programme in 2019 and 2018, respectively. Rogert Sorí and Marta Vázquez are supported by the Xunta de Galicia (grant nos. ED481B 2019/070 and ED481B 2018/062), respectively. Margarida L. R. Liberado and Milica Stojanovic are supported by the project "Weather Extremes in the Euro Atlantic Region: Assessment and Impacts-WExAtlantic" (grant no. PTDC/CTA-MET/29233/2017). This work is part of the LAGRIMA project (grant no. RTI2018-095772-B-I00) funded by Ministerio de Ciencia, Innovación y Universidades, Spain. Partial support was also obtained from the Xunta de Galicia under the project "Programa de Consolidación e Estructuración de Unidades de Investigación Competitivas (Grupos de Referen-

cia Competitiva)" (grant no. ED431C 2017/64-GRC). All the authors acknowledge the support of Alex Ramos from the Instituto Dom Luis for interpreting the CWT computations.

*Financial support.* This research has been supported by the European Association for Territorial Cooperation, Galicia Northern Portugal (AECP, GNP) through the IACOBUS programme in 2019 (grant nos. 96 and 262); the Xunta de Galicia (grant no. ED431C 2017/64-GRC); the RISC_ML project under the INTERREG España-Portugal programme (grant no. POCTEP; 2014-2020); the Ministerio de Ciencia, Innovación y Universidades (grant no. LAGRIMA RTI2018-095772-B-I00); and the Fundação para a Ciência e a Tecnologia (FCT), Portugal (grant no. FCT/DFRH/SFRH/BPD/84328/2012). TS35

*Review statement.* This paper was edited by Chris Reason and reviewed by two anonymous referees.

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

## Remarks from the language copy-editor

## Remarks from the typesetter

**TS31** Please provide a statement on how your underlying research data can be accessed. If the data are not publicly accessible, a detailed explanation of why this is the case is required. The best way to provide access to data is by depositing them (as well as related metadata) in reliable public data repositories, assigning digital object identifiers (DOIs), and properly citing data sets as individual contributions. Please indicate if different data sets are deposited in different repositories or if data from a third party were used. Additionally, please provide a reference list entry including creators, title, and date of last access. If no DOI

is available, assets can be linked through persistent URLs to the data set itself (not to the repositories' home page). This is not seen as best practice and the persistence of the URL must be secured.

TS32 Please note that the section "Author contributions" is mandatory.

TS33 Declaration of all potential conflicts of interest is required by us as this is an integral aspect of a transparent record of scientific work. If there are possible conflicts of interest, please state what competing interests are relevant to your work.

TS34 Please confirm.

TS35 Please note that there is a discrepancy between funding information provided by you in the acknowledgements and the funding information you indicated during manuscript registration, which we used to create this section. Please double-check your acknowledgements to see whether repeated information can be removed from the acknowledgements or changed accordingly. If further funders should be added to this section, please provide the funder names and the grant numbers. Thanks.

TS36 Please provide last access date.

TS37 Please provid last access date.

TS38 Please provide last access date.

TS39 Please provide volume and check page range.

TS40 The reference of Blanco-Durán et al. (2012) is not mentioned in this paper. Please check.

TS41 Please provide title.

TS42 Please provide all author names.

TS43 Is this García-Herrera? Please check.

TS44 Please provide page range or article number.

TS45 Please provide place of publication.

TS46 Please provide volume and check page range.

TS47 Is this Martín-Vide? Please check.

TS48 Please provide full page range.

TS49 Please provide publisher and place of publication.

TS50 Please provide volume and check page range.

TS51 Please provide ldast access date.

TS52 Please provide last access date.

TS53 Please provide last access date.

TS54 Is this Rodríguez-Puebla? Please check.

TS55 Please provide full page range.

TS56 Is this C. G. Torrence? Please check.

TS57 Please check page range.

TS58 Please check DOI number.

TS59 Please check page range.

TS60 Please provide full page range.

TS61 Please provide journal name and volume and check article number.