# Peer review of "Hydrometeorological droughts in the Miño-Limia-Sil hydrographic demarcation (NW Iberian Peninsula): The role of atmospheric drivers"

_Natural Hazards and Earth System Sciences, 2019_

## Referee Comment (RC1) · Anonymous Referee #1 · 29 Oct 2019

General comments:

In this paper, meteorological droughts affecting the Miño-Limia-Sil Hydrographic Demarcation during the period of 1980–2017 are identified and assessed using the one month SPEI index (and, in some cases, some other temporal scales). In this way, the problems associated with droughts, their origin and their impacts are analyzed at a regional level. It is shown that the driest/wettest conditions occur under some particular Circulation Weather Types. In addition, some teleconnection patterns seem to favor more/less frequent dry conditions with different temporal scales. Finally, soil moisture and river stream flows are also related to drier or wetter conditions in previous momths.

[Figure]

My general impression is that many of the presented results are not new, since most of the results shown in this document are in agreement with those presented in previous papers on droughts in the northwest of the Iberian Peninsula. Probably, the most original aspect of this paper is that it shows that the methods previously used in the NW Iberian Peninsula can be used successfully in a much smaller region, such as MLSHD. Anyhow, I think that this paper is acceptable for publication with some revisions and clarification.

Specific comments:

In general, the indices used in the paper are described very diffusely and only some random information about them is provided to the reader (of course, references are provided on how those indices are defined. But further explanations would help to better understand what is presented in the paper). I think more information has to be provided on how those indices are calculated and what they represent.

For example, why doesn't the 'materials and methods' section include a brief definition of SPEI and SSI and the way they are computed? Equations 1 and 2 define Eto (E1) and Rn2(S), but the text does not include the definition of some of the indices most frequently used in the paper (SPEI, SSI, severity...) or, for example, the way in which the statistical sifnificance of the wavelet coherence is calculated.

Some information about the data is missing as well. Are the Miño and Limia rivers discharge series affected by reservoirs or by any other human regulation activity? This information should be included in the description of the data and considered in the discussion of the results. In the conclusions section, it is explained that the classification of the CWTs is daily. I think this information should be included in the description of the data.

The Atlantic Multidecadal Oscillation is known to be characterized by a period that varies from approximately 40 to 80 years (even if a band of 20-30 years is accepted, the period studied in this paper does not even include two complete cycles of such

an AMO oscillation). Therefore, it is very difficult to assess the impact of AMO with a study period of less than 40 years. If the maximum coherence period between AMO and SPEI1 is 6 years (figure 11) then it is not addequate to describe it as multidecadal coherent oscillation. My opinion is that the analysis of the impact of the AMO is too noisy (and not multidecadal enough) and that it should not be included in this paper.

Other suggestions, corrections and typos:

Abstract: Is it acceptable to define acronyms in the abstract? If so, MLSHD is defined for the first time in the abstract and it is not needed to re-define it in the Introduction (P3-L6)

Abstract-line 19: 'The results showed that atmospheric circulation from the southwest, west, and northwest were directly related to dry and wet conditions': To both dry and wet conditions?

Abstract-line 22: 'the major teleconnection atmospheric patterns': change to "the major atmospheric teleconnection patterns"

Page 3-Line 15: Delete '(Figure 1)': the MLSHD and figure 1 are already referenced a few lines before.

P4-L15: define the variables in ec. 1

P4-L17: SPEI is said to be a multiscale index, but it is not clear what is the advantage of this multiscale character. In fact, most of the paper deals only with SPEI1. Please, describe those advantages.

P4-L21: 1954-2014: 61 years, thus about six decades, not five.

P5-L13: ERA-Interim reanalysis for the period 1979-2017. If not, the period 1980-2017 cannot be set for all the analysis in this study (P6-L13).

P5-L18 to L23: a better explanation of what 'pure' or hybrid' WTs are (and about the implications of WTs being 'pure' or 'hybrid') would be appreciated. A mathematical

definition based on poorly defined parameters is not enough. Any mention to those 16 'hybrid' circulations could be removed from the main text since they are not mentioned in any other section of the paper.

P7-L12: '...occurred between December and February...': ?

P8-L6: '...the length of these episodes increased after 2003': Not the length, it is the frequency of long episodes what is higher. Long episodes can be found in 1988-1993 & 1980-1983, but after 2003 they are more frequent.

P8-L16-20: the trends described in these lines are far from being statistically singificant (p values of 0.26, 0.52 and 0.26!!). Thus, it is difficult for me to understand why authors make so much emphasys in these trends.

P9-L4: What do the numbers included as 'severity' in table 3 mean? How are these 'severity' values calculated? Same comment is applicable in table 2 and P8-L18. The definition of 'severity' and how it is calculated should be included in section 2.

P11-L1: Pressure values in figure 4 are very small and difficult to read. The caption of this figure could include that reddish (blueish) isolines represent high (low) pressure values.

P11-L18: Figure 5 caption: what do 'X's in the figure mean? I guess they represent not significant correlations, but it should be stated in the caption. What does the size of the circles in this figure mean? Is it just proportional to the value of the corralation? That information should be in the caption.

P11-L22: moNths. The 'N' is missing

P12-L8-10: The description of what is shown in figure 7 included in the main text does not coincide with the caption of the figure. The caption seems to be wrong. Please, revise it.

P13-L3: Figure 7: Why are percentages negative? What WT is the one with a positive

percentage? Its color is not included in the WT color table. Please, revise the caption and the main text and include this information.

P13-L10: Both the main text and the caption of Figure 8 should explain what the authors mean when they talk about the onset, the peak and the termination of the drought episodes. I guess the "onset" is the month in which the episode begins, the "peak" is the month in which the episodes reach the highest value of SPEI1 and the 'termination' is the month after the month in which the episode ends. But these ideas are not clear in the main text or in the caption.

P19-Figure 10 caption: Correlations shown in figure 10a are obtained from monthly series? What are the 'X's in figure 10b? Confidence level? Figures 10 and 11 Captions: SPEI1 is enough, delete '...the 1-mo Standarised Precipitation-Evapotranspiration Index...' (idem in figure 11 caption)

P19-L25 to P20-L2: The interpretation of figure 11 is already included in the figure caption. It could be deleted from the main text.

P21-L19: Should it be '...increased with the TEMPORAL SCALE OF SPEI...'

P21-L20: 'Figure 12b and c...' should be a new paragraph

P21-L31: '...basin features and water REGULATION...' this point is very important in the interpretation of the results. Nowhere in the paper it is said wether the streams were regulated or natural. I guess they are regultaed and, thus, it is difficult to obtain clear interpretations from them.

P22-L13-14: I do not understand very well what authors mean with their sentence 'The results revealed the frequency of the WTs prone...' Please, revise.

P23-L3-8: As I already said, I do not think any conclusion can be obtained about the influence of AMO.

P23-L10-14: Conclusions about soil moisture are sound, but results about river streamflows are much more duvious since those flows are most probably regulated.

---

## Referee Comment (RC2) · Anonymous Referee #2 · 6 Nov 2019

Review of the manuscript nhess-2019-314, entitled "Meteorological drought in the Miño-Limia-Sil hydrographic demarcation: The role of atmospheric drivers", submitted by Rogert Sorí et al. for possible publication in Natural Hazards and Earth System Sciences.

Recommendation Sorí et al. assessed drought characteristics in the Miño-Limia-Sil basin (NW Iberia) from 1980 to 2017 using the Standardized Precipitation Evapotranspiration Index (SPEI). The temporal variability of drought metrics was linked to changes in the dominant weather types and atmospheric circulation patterns in the region. Weather types in the study domain were classified using an automated version

of the standard Lamb weather types' scheme. The study is interesting from the climatological point of view and falls within the scope of NHESS. The level of innovation of this work is fair/reasonable. Albeit with the availability of several studies, which assessed drought characteristics in the IP (most of them are referenced herein), this study applied well-established/existing methods in a standard fashion for a cross-boundary basin in the IP. The manuscript is generally well-written. However, the manuscript cannot be accepted for publication in its current version. Some methodological issues should be clarified. The structure of the manuscript should be refined. My major and specific comments are listed below:

I. Major comments - I want to point out one smaller aspect that deals with the general readability of the text. The readability is sometimes hampered by the use of various abbreviations (e.g. P, Y, M, MSRB, LRB, MLSHD, NWIP, VIMF, etc). If the reader is not very familiar with the terms and abbreviations, it is hard to follow. I would suggest using only those abbreviations that are reoccurring and central for the topic and trying to avoid others. This would strongly improve readability. - In the methods section, the authors need to clarify the final number of weather types retained in their work. Have they applied any regrouping/reclassification of hybrid types? What are the stopping criteria of their classification scheme? What statistical significance criteria are applied? How robust and reproducible are the results? What guided the reduction and grouping? Can you illustrate what the types represent, via typical flow or MSLP patterns? Also, a map illustrating the 16 points used for weather type classification should be included. - I am wondering why the authors have not used some Mediterranean specific indices, such the WeMO and MO. Also, I am wondering why the authors did not consider SST as a driver of drought variability in their domain. This can be implemented using El Niño 3.4 index (SST anomalies in the central Pacific), El Niño 1 + 2 index (SST anomalies in the eastern Pacific) or SST anomalies in the tropical Atlantic. I am aware that the manuscript focuses mainly on specific atmospheric drivers, but at least in the discussion, the authors need to highlight the possible role of SST warming in drought reinforcement. See, for example: https://doi.org/10.1175/JCLI-D-11-00296.1 - The authors should discuss their results in the context of some earlier studies whose results contradict the findings of the current study (particularly with respect to the significant role of NAO in drought variability in Europe in general and the IP in particular). See for example: https://doi.org/10.1007/s10584-007-9285-9. 10.1007/978-94-007-1372-7_3 - The authors focused mainly on the temporal variability of drought and its connections with WTs and climate teleconnections. However, I have not seen any attempt to show the varying spatial response of drought to these drivers. The reasonable spatial resolution ($0.1°$) of E-OBS allows for a reliable assessment of the spatial variability of drought in response to the different atmospheric configurations. The authors indicate in the abstract "We concluded that regional patterns of land-use change and moisture recycling are important to consider in explaining runoff change, integrating land and water management, and informing water governance". I think decision-makers and water resources planners in any catchment seek for detailed information on the spatial variability of droughts so that they can adopt integrated policies and strategies for managing their catchment, taking into consideration the different conditions at both upstream and downstream. - In the Introduction, the authors lack the opportunity to comprehensively provide evidence on the hydrological, environmental and socioeconomic importance of MLSHD in the IP. - A short description of the study domain, highlighting its main physiological, climatic and hydrological settings is needed. Section 3.1 is misplaced in the results section and should be forwarded to a new subsection called "study area". - A justification of the selection of the (-0.84) as a threshold for defining drought events is needed. - In the methodology, the authors should clarify how the different drought metrics were computed? How were the trend and its statistical significance assessed? Have they accounted for the possible presence of serial correlation in the series? - The authors should clarify the rationale behind constraining their study from 1980, while E-OBS dataset extends back to the 1950s (probably not for all climate variables). - I would recommend adding a new figure, in which the authors compare the accumulated SPEI values corresponding to each WT. - Prior to calculating correlation, it is important to detrend the series of the frequency of WT. - In the discussion, the authors should refer

[Figure]

to the role of zonal and meridional circulation in drought characteristics (in the context of the directional WTs). - It is recommended to add a table, which summarizes trends of the 10 main WTs over the study period, their statistical significance and compares those with the trends observed for the different drought metrics (e.g. drought, severity, and occurrence). - HS method for calculating PET is a temperature-based method, which is more suitable for arid and semi-arid regions (not humid regions like the study domain). Have the authors tested the performance/accuracy of this method in their region? Several studies reported a less performance of temperature-based methods in assessing PET in humid climates. - Given the limited area of the study domain, the spatial resolution of SMroot data seems to be coarse ($0.25° \times 0.25°$) to provide a reliable assessment of the response of soil moisture to precipitation deficit. Also, in humid climates like those of the study area, the response of soil moisture to accumulated precipitation deficit is more pronounced at longer time scales (not 1-month time scale). The persistence of negative soil moisture anomalies is expected to be higher when there is a cumulative long-term decrease in the amount of precipitation. This aspect should be discussed thoroughly. - Describe all symbols given in Eq. 1. - Which index exactly of the NAO, as well as ENSO, was used? Please, be more specific. There are different indices for quantifying each of them. - Section 2.5 should be placed earlier in the materials and methods section (before the description of drought calculation). - The role of aerodynamic components in drought evolution should be discussed, given that these influences are not considered in HS method.

II. Minor comments - Title: It is recommended to indicate the location of the study domain (i.e. NW Iberia), as the majority of the NHESS readers are not familiar with the study basin. Also, it is important to include "hydrological droughts" in the title. P1 - L14 and other parts of the ms: "period of" <> "period". - L16 and other parts of the ms: "mo" <> "month". - L17: For a study that covers 38 years, the use of the term "historically" is misleading; please, define the confidence interval at which the significance was assessed. - L18: "different" <> "the different". - L19: Based on which scheme this classification was made? The abstract should stand alone based on this basic information. - L20: "were directly related to dry and wet conditions" This statement is vague, with no clear phrasing. It does not make a clear conclusion on whether these weather types are favoring for above-normal or below-normal precipitation. - L25: Please, define the rainy season. - L27 and other parts of the ms: "1 y" <> "1 year". - It is unclear how meteorological droughts assessed at 1-month time scale can be linked with land use changes (which almost occur at a coarse temporal scale".

P2 - L5: Please, give some examples of these thermodynamic factors (e.g. wind speed, air pressure). - L15: Delete "e.g." - L19: "the precipitation" <> "precipitation". - L25: "land" Do you mean air temperature? LST has a different conception and is mostly assessed using remote sensing products (e.g. MODIS, AVHRR), which are only available for the most recent decades. - L28: The study of Vicente-Serrano et al. (2011) does not provide any assessment of future projections of precipitation P3 - L8: "a homogeneous region in terms of the total P variance over the IP". This statement should be elaborated thoroughly. P4 - L25: "A drought episode was considered to occur when the SPEI at the temporal scale of 1 mo fell below zero, reached a value of at least -0.84, and later returned to positive values". This definition should be made simpler.

P5 - L1: "Results" <> "Results and discussion". - L9: Language and style should be revised. - L12: "modulate" <> "impact". - It is unclear why the classification of weather types is only restricted to the period 1989-2017. - L15: Please, define this spatial window. - For classifying weather types, the authors should clarify how SF, WF, ZS, ZW, F, and Z were computed?

P6 - L20: "from daily values" <> "aggregated from daily values". - L21: The name of the station "Albufeira Do Alto" does not fit with that labeled in Figure 1.

P7 - L5: "the annual cycle" <> "the year"; "western" <> "the western". - L20: What is the difference between "extensive" and "intense"? Do you refer to drought duration and severity?

P8 - L30: "for in" <> "for". - The acronyms "WTs" and "CWTs" are used interchangeably

in the text. - P23 (L10): "the soil" <> "soil". - P23 (L10): "the moisture" <> "moisture". - This work emphasized that drought did not respond linearly to most of the dominant circulation patterns in this region (apart from SCAN, AO) at 1-month timescale (Figure 10a). This finding should be discussed thoroughly in the text and linked with available literature.

Tables - Table 1: There is a refinement of the drought categories of Agnew (mild drought is masked with another category). - Table 2: Trends in SPEI values should be expressed in z-units/year.

Figures - Figure 1: In the legend and caption, "rivers" <> "streams". The negative symbol corresponding to the longitudes should be deleted, given that the direction "W" is already included. It is important to include a distribution of the meteorological stations whose data were used for SPEI calculation. - Figure 3: how were drought episodes defined? Have you applied n consecutive months with SPEI <-0.84? - Figure 4: I would recommend using the anomalies (not the actual values) of SLP corresponding to the different WTs. This will facilitate defining the positive and negative centers of action that control air advection at the surface. - Figure 5: The use of the symbol "x" should be described. The use of the legend in a vertical form is confusing, given that all WTs at the top of the panel show a negative correlation (shown in blue). I would recommend reversing the legend so that negative values of correlation are shown at the top, while negative correlations are illustrated below. Why the authors did not use a portrait diagram showing the interpolated surface of the correlation coefficient, with some contour lines to show the significance of the correlation? This will facilitate the readability of the figure. - Figure 6: I would recommend plotting the events at the x-axis, while the stacked bars show the contribution of each WT to such events. This will deliver the message clearer. - Figure 7: why the percentages are given in negative? To which WT refers the "red" color? I would recommend adding a column to the three drought categories, which refers to wet conditions (i.e. SPEI values >0). This contrast can show interesting results about the role of each WT during dry vs. wet conditions.

---

## Author Comment (AC1) · 31 Dec 2019

General comments: In this paper, meteorological droughts affecting the Miño-Limia-Sil Hydrographic Demarcation during the period of 1980–2017 are identified and assessed using the one month SPEI index (and, in some cases, some other temporal scales). In this way, the problems associated with droughts, their origin and their impacts are analyzed at a regional level. It is shown that the driest/wettest conditions occur under some particular Circulation Weather Types. In addition, some teleconnection patterns seem to favor more/less frequent dry conditions with different temporal scales. Finally, soil moisture and river stream flows are also related to drier or wetter conditions in previous months. My general impression is that many of the presented results are not new, since most of the results shown in this document are in agreement with those presented in previous papers on droughts in the northwest of the Iberian Peninsula. Probably, the most original aspect of this paper is that it shows that the methods previously used in the NW Iberian Peninsula can be used successfully in a much smaller region, such as MLSHD. Anyhow, I think that this paper is acceptable for publication with some revisions and clarification.

Specific comments: In general, the indices used in the paper are described very diffusely and only some random information about them is provided to the reader (of course, references are provided on how those indices are defined. But further explanations would help to better understand what is presented in the paper). I think more information has to be provided on how those indices are calculated and what they represent. For example, why doesn't the 'materials and methods' section include a brief definition of SPEI and SSI and the way they are computed? Equations 1 and 2 define Eto (E1) and Rn2(S), but the text does not include the definition of some of the indices most frequently used in the paper (SPEI, SSI, severity...) or, for example, the way in which the statistical sifnificance of the wavelet coherence is calculated. Some information about the data is missing as well. Are the Miño and Limia rivers discharge series affected by reservoirs or by any other human regulation activity? This information should be included in the description of the data and considered in the discussion of the results. In the conclusions section, it is explained that the classification of the CWTs is daily. I think this information should be included in the description of the data.

*We appreciate your advices to improve the manuscript. The methodology section was improved in order to provide a better description of the indices utilized, the calculation of drought indicators, the computation of WTs and the Wavelet coherence significance.*

The Atlantic Multidecadal Oscillation is known to be characterized by a period that varies from approximately 40 to 80 years (even if a band of 20-30 years is accepted, the period studied in this paper does not even include two complete cycles of such C2 NHESSD Interactive comment Printer-friendly version Discussion paper an AMO oscillation). Therefore, it is very difficult to assess the impact of AMO with a study period of less than 40 years. If the maximum coherence period between AMO and SPEI1 is 6 years (figure 11) then it is not addequate to describe it as multidecadal coherent oscillation. My opinion is that the analysis of the impact of the AMO is too noisy (and not multidecadal enough) and that it should not be included in this paper.

*We acknowledge your advice! We agree and consequently the AMO analysis and discussion was removed from the text.*

Other suggestions, corrections and typos:

Abstract: Is it acceptable to define acronyms in the abstract? If so, MLSHD is defined for the first time in the abstract and it is not needed to re-define it in the Introduction (P3-L6)

*The acronyms were deleted from the abstract.*

Abstract-line 19: 'The results showed that atmospheric circulation from the southwest, west, and northwest were directly related to dry and wet conditions': To both dry and wet conditions?

*Than you. This sentence was modified:*

*"The results showed that atmospheric circulation from the south-east/west, east/west, and north-east/west were directly related to dry/wet conditions in the Miño-Limia-Sil Hydrographic Demarcation during the entire climatological year"*

Abstract-line 22: 'the major teleconnection atmospheric patterns': change to "the major atmospheric teleconnection patterns"

*Changed*

Page 3-Line 15: Delete '(Figure 1)': the MLSHD and figure 1 are already referenced a few lines before.

*Deleted*

P4-L15: define the variables in ec. 1

*You are right They are now defined in the manuscript*

P4-L17: SPEI is said to be a multiscale index, but it is not clear what is the advantage of this multiscale character. In fact, most of the paper deals only with SPEI1.

*Yes, we mostly utilized the SPEI1 to identify meteorological droughts, which we believe are the best related with different WTs. Besides the meteorological drought can be perceived as the initial cause of other types of drought. The multiscalar advantage permit to assess whether this accumulated dry/wet conditions impact other steps of the hydrological cycle; in this case the soil moisture content and the runoff. The explanation was improved in the manuscript.*

P4-L21: 1954-2014: 61 years, thus about six decades, not five.

*Changed*

P5-L13: ERA-Interim reanalysis for the period 1979-2017. If not, the period 1980-2017 cannot be set for all the analysis in this study (P6-L13).

*Thank you, it was corrected. But the correct study period is 1980-2017 in order to fit a temporal scale with complete data of all variables (in this case the river discharge was complete from 1980 to 2017).*

P5-L18 to L23: a better explanation of what 'pure' or hybrid' WTs are (and about the implications of WTs being 'pure' or 'hybrid') would be appreciated. A mathematical definition based on poorly defined parameters is not enough. Any mention to those 16 'hybrid' circulations could be removed from the main text since they are not mentioned in any other section of the paper.

*The description of the methodology referred to the weather type computation was modified in order to provide further details. The difference between pure and hybrid types was introduced into the text as well as a description of how both types are considered in the percentage contribution.*

*"According to the methodology developed by Trigo and Da Camara (2000), 10 different "pure" WTs can be identified, namely Northeastern (NE), Eastern (E), Southeastern (SE), Northwestern (NW), Western (W), Southwestern (SW), North (N), South (S), Anticyclonic (A), and Cyclonic (C). Pure directional WTs (NE, E, SE, NW, W, SW, N, S) were those showing $|Z| < F$ with the direction defined by tan-1 (WF/SF) (180° added if WF is positive).If $|Z| > 2F$, then the circulation would be considered C (if $Z > 0$) or A (if $Z < 0$). As not all the circulation patterns could be associated with a pure (directional/cyclonic/anticyclonic) type, 16 hybrid circulations were defined as a combination of A and C circulation with directional WTs. In this case, $F < |Z| < 2F$.*

*The methodology here described is able to daily identify the weather pattern (from the 26 listed before) presented over the area of study. From this daily information, and in order to study the WTs influence on monthly SPEI series, the monthly frequency of occurrence for every pure WTs is computed for the period 1979-2017. In the frequency computation, the 26 WTs are regrouped in the 10 "pure" ones. This procedure was realized following the same approach applied in Trigo and DaCamara (2000) in which the hybrid types were included into the corresponding pure WTs with a weight of 0.5, being the 10 final number of WTs analysed in this study."*

P7-L12: '...occurred between December and February...': ?

*It was changed for "from December to February"*

P8-L6: '...the length of these episodes increased after 2003': Not the length, it is the frequency of long episodes what is higher. Long episodes can be found in 1988-1993 & 1980-1983, but after 2003 they are more frequent.

*Thank you. It has been corrected on the text.*

P8-L16-20: the trends described in these lines are far from being statistically significant (p values of 0.26, 0.52 and 0.26!!). Thus, it is difficult for me to understand why authors make so much emphasys in these trends.

*The paragraph was modified according to the reviewer comment and this result was removed from the abstract as it not represents a relevant conclusion.*

P9-L4: What do the numbers included as 'severity' in table 3 mean? How are these 'severity' values calculated? Same comment is applicable in table 2 and P8-L18. The definition of 'severity' and how it is calculated should be included in section 2.

*In section 2 as added the explanation:*

*-          The duration is computed as the sum of all months from the onset with negative values, the peak is the month in which the episodes reach the highest value of SPEI1, and the severity is calculated as the sum of all SPEI values (in absolute values) during the episode.*

P11-L1: Pressure values in figure 4 are very small and difficult to read. The caption of this figure could include that reddish (blueish) isolines represent high (low) pressure values.

*The figure and caption was changed to solve this!*

P11-L18: Figure 5 caption: what do 'X's in the figure mean? I guess they represent not significant correlations, but it should be stated in the caption. What does the size of the circles in this figure

mean? Is it just proportional to the value of the correlation? That information should be in the caption.

*The information was included in the caption:*

*"The x's in the figure represent not statistical significant correlations at p < 0.05. The size for the circles is proportional to the correlation values"*

P11-L22: moNths. The 'N' is missing

*Thank you. It has been corrected*

P12-L8-10: The description of what is shown in figure 7 included in the main text does not coincide with the caption of the figure. The caption seems to be wrong. Please, revise it.

*The caption was wrong and it has been modified:*

*''Monthly percentage of occurrence for every WT associated with moderate, severe and extreme dry conditions. The red bars represent the number of times that each month was affected by each drought category''*

P13-L3: Figure 7: Why are percentages negative? What WT is the one with a positive paper percentage? Its color is not included in the WT color table. Please, revise the caption and the main text and include this information.

*The negative sing is a mistake in the figure, it was deleted. The original positive value (red bars) does not represent any weather type, just the number of times that each month was under different drought categories. It has been clarified in the text and in the caption.*

P13-L10: Both the main text and the caption of Figure 8 should explain what the authors mean when they talk about the onset, the peak and the termination of the drought episodes. I guess the "onset" is the month in which the episode begins, the "peak" is the month in which the episodes reach the highest value of SPEI1 and the 'termination' is the month after the month in which the episode ends. But these ideas are not clear in the main text or in the caption.

*The description of onset, termination, and peak and other important terms were now introduced and best explained into the text accordingly with your suggestion. And yes, in this table the **onset** and the **end** represent the first and last month of the episode and the **peak** is the highest SPEI1 value reached in the episode.*

P19-Figure 10 caption: Correlations shown in figure 10a are obtained from monthly series? What are the 'X's in figure 10b? Confidence level?

*Monthly time series were used to obtain correlations in Figure 10. The x's represent not significant correlation with 95% confidence level. This information was included in the caption.*

Figures 10 and 11 Captions: SPEI1 is enough, delete '...the 1-mo Standarised Precipitation-Evapotranspiration Index...' (idem in figure 11 caption)

*Changed*

P19-L25 to P20-L2: The interpretation of figure 11 is already included in the figure caption. It could be deleted from the main text.

*It was deleted form the text.*

P21-L19: Should it be '...increased with the TEMPORAL SCALE OF SPEI...'

*Changed*

P21-L20: 'Figure 12b and c...' should be a new paragraph

Changed

P21-L31: '...basin features and water REGULATION...' this point is very important in the interpretation of the results. Nowhere in the paper it is said wether the streams were regulated or natural. I guess they are regulated and, thus, it is difficult to obtain clear interpretations from them.

*We understand your concern and we confirmed that on the Spanish part of the MLSHD exist more than 2000 dams (according to the Hydrographic Confederation: [https://www.chminosil.es/es/](https://www.chminosil.es/es/)). In the Portuguese part are also several dams and hydroelectric stations. For this reason, we removed the analysis of the SSI. Although, Añel et al., (2014) found the dams in the Miño-Sil river basin had no influence on the natural river flows over the period 1978-2012 and based on this report we performed the initial study with the SSI. ([http://catedranaturgy.webs4.uvigo.es/content/Workingpaper.pdf](http://catedranaturgy.webs4.uvigo.es/content/Workingpaper.pdf))*

*The same analysis was performed but utilizing datasets of runoff at a resolution of ~ 4 km to compute the Standardised Runoff Index (SRI). An explanation of this index was added in section 2.*

P22-L13-14: I do not understand very well what authors mean with their sentence 'The results revealed the frequency of the WTs prone...' Please, revise.

*The sentence was changed*

P23-L3-8: As I already said, I do not think any conclusion can be obtained about the influence of AMO.

*After considering your concern, we agree, and the analysis of the AMO was removed from the manuscript.*

P23-L10-14: Conclusions about soil moisture are sound, but results about river flows are much more duvious since those flows are most probably regulated.

*Due to the resolution of the soil moisture data (0.25º) we decided to utilize another source of soil moisture. In this case as commented before we utilize the soil moisture from Terraclimate, which has a resolution of ~ 4km and is consequently more representative for the region. We used here the Standardized Soil Moisture Index (SSMI) (Haoand AghaKouchak, 2013)*

---

## Author Comment (AC2) · 31 Dec 2019

Review of the manuscript nhess-2019-314, entitled "Meteorological drought in the Miño-Limia-Sil hydrographic demarcation: The role of atmospheric drivers", submitted by Rogert Sorí et al. for possible publication in Natural Hazards and Earth System Sciences. Recommendation Sorí et al. assessed drought characteristics in the Miño-Limia-Sil basin (NW Iberia) from 1980 to 2017 using the Standardized Precipitation Evapotranspiration Index (SPEI). The temporal variability of drought metrics was linked to changes in the dominant weather types and atmospheric circulation patterns in the region. Weather types in the study domain were classified using an automated version of the standard Lamb weather types' scheme. The study is interesting from the climatological point of view and falls within the scope of NHESS. The level of innovation of this work is fair/reasonable. Albeit with the availability of several studies, which assessed drought characteristics in the IP (most of them are referenced herein), this study applied well-established/existing methods in a standard fashion for a cross-boundary basin in the IP. The manuscript is generally well-written. However, the manuscript cannot be accepted for publication in its current version. Some methodological issues should be clarified. The structure of the manuscript should be refined. My major and specific comments are listed below:

I.      Major comments

I want to point out one smaller aspect that deals with the general readability of the text. The readability is sometimes hampered by the use of various abbreviations (e.g. P, Y, M, MSRB, LRB, MLSHD, NWIP, VIMF, etc). If the reader is not very familiar with the terms and abbreviations, it is hard to follow. I would suggest using only those abbreviations that are reoccurring and central for the topic and trying to avoid others. This would strongly improve readability.

*Following your advice some of the abbreviations were removed such as "y", "mo", NWIP, MSRB, LRB?*

In the methods section, the authors need to clarify the final number of weather types retained in their work. Have they applied any regrouping/reclassification of hybrid types? What are the stopping criteria of their classification scheme? What statistical significance criteria are applied? How robust and reproducible are the results? What guided the reduction and grouping? Can you illustrate what the types represent, via typical flow or MSLP patterns? Also, a map illustrating the 16 points used for weather type classification should be included.

*The total number of types was included in the study but, as suggested by the reviewer a regrouping of the hybrid types was realized in order to obtain the monthly percentage for every weather types. This and other issues addressed by the reviewer where added into the text to explain better the WTs computation process. The MSLP associated with every WTs is presented in Figure 4 and the points used in the computation are represented in Figure 1 (right panel). Transcription is presented below:*

*"The methodology here described is able to daily identify the weather pattern (from the 26 listed before) presented over the area of study. From this daily information, and in order to study the WTs influence on monthly SPEI series, the monthly frequency of occurrence for every pure WTs is computed for the period 1979-2017. In the frequency computation, the 26 WTs are regrouped in the 10 "pure" ones. This procedure was done following the same approach applied in Trigo and DaCamara (2000) in which the hybrid types count equally as a half occurrence to each of their pure types, being the 10 final number of WTs analysed in this study."*

*Regarding stopping criteria and statistically significance, it has been applied in other studies a 'stopping rule' to determine, for example, at what point to stop adding predictors (WTs) to obtain*

*the relative (%) contribution to total monthly Iberian rainfall by each WT. However, is not our case. However, as in other previous studies we distributed the few cases (<2%) with unclassified situations among the 26 classes. The unclassified situations occur when the module of the total shear vorticity (Z) is less than the standard deviation of Z/2 and simultaneously the total flow (F) is less than the square root of the sum of squared westerly and southerly flows.*

*The types represent the MSLP. We considered the circulation associated to each WT to explain the role of directional flows. Several studies explain how to obtain the WT through the methodology here utilized, so, the results are perfectly reproducible.*

I am wondering why the authors have not used some Mediterranean specific indices, such the WeMO and MO. Also, I am wondering why the authors did not consider SST as a driver of drought variability in their domain. This can be implemented using El Niño 3.4 index (SST anomalies in the central Pacific), El Niño 1 + 2 index (SST anomalies in the eastern Pacific) or SST anomalies in the tropical Atlantic. I am aware that the manuscript focuses mainly on specific atmospheric drivers, but at least in the discussion, the authors need to highlight the possible role of SST warming in drought reinforcement. See, for example: https://doi.org/10.1175/JCLI-D-11-00296.1

*Thank you for your recommendation. These Index were also investigated and no significant correlation was obtained for any of El Niño Index (r values from 0.05 to 0.25 with the SPEI6 and TNA (r < 0.07). This information is included in the manuscript. The Index BEST already introduce information of SST of the region 3.4 and also takes into account the SOI (the pressure difference between Tahiti and Darwin), to give a better representation of the phenomenon. It has been argued in the manuscript.*

*The analysis for the WeMO index was incorporated and removed the AMO.*

The authors should discuss their results in the context of some earlier studies whose results contradict the findings of the current study (particularly with respect to the significant role of NAO in drought variability in Europe in general and the IP in particular). See for example: https://doi.org/10.1007/s10584-007-9285-9. 10.1007/978-94-007-1372-7_3

*We improve the discussion considering this advice. The link seems to be wrong.*

The authors focused mainly on the temporal variability of drought and its connections with WTs and climate teleconnections. However, I have not seen any attempt to show the varying spatial response of drought to these drivers. The reasonable spatial resolution (0.1º) of E-OBS allows for a reliable assessment of the spatial variability of drought in response to the different atmospheric configurations. The authors indicate in the abstract "We concluded that regional patterns of land-use change and moisture recycling are important to consider in explaining runoff change, integrating land and water management, and informing water governance". I think decision-makers and water resources planners in any catchment seek for detailed information on the spatial variability of droughts so that they can adopt integrated policies and strategies for managing their catchment, taking into consideration the different conditions at both upstream and downstream.

*To assess this, we decided to perform an EOF analysis for those months of the MLSHD when the average SPEI was ≤ - 0.84. In this analysis the PCA of the first EOF were correlated with each series of the WTs for the same chosen months. The first 6 EOF explain more than the 95% of the total*

*explained variance, and the two first around the 80%. Figures (EOF & PCA) has been added to the manuscript.*

In the Introduction, the authors lack the opportunity to comprehensively provide evidence on the hydrological, environmental and socioeconomic importance of MLSHD in the IP. A short description of the study domain, highlighting its main physiological, climatic and hydrological settings is needed. Section 3.1 is misplaced in the results section and should be forwarded to a new subsection called "study area".

*Thank you for your advice. The Introduction has been improved by adding new information about its hydrological, environmental and socioeconomic characteristic of importance. The section 3.1 was also moved to the Introduction section in order to provide a more compressive information about the climate of the region.*

A justification of the selection of the (-0.84) as a threshold for defining drought events is needed.

*We improved this in section 2 as follows:*

*We avoid considering that any precipitation below the mean constitutes a drought. Therefore, the classification of drought categories according to SPI values proposed by Agnew (2000) (Table 1) was utilised in this study. Other authors have also employed this classification for investigating drought in the IP (e.g. Vicente-Serrano et al., 2006; Páscoa et al., 2017). This classification despite to be pre-established is built by probability classes rather than magnitudes of the SPI, and is therefore suggested as a more rational approach, with a most noticeable effect at the demarcation of mild and moderate droughts (Agnew, 2000). Then a drought episode was considered to occur (onset) when the SPEI at the temporal scale of 1 month fell below zero, reached a value of at least -0.84, and later returned to positive values. The threshold of -0.84 corresponds to 5% probabilities, whereby drought is only expected 2 years in 10, which reduce the incidence of mild meteorological droughts.*

In the methodology, the authors should clarify how the different drought metrics were computed? How were the trend and its statistical significance assessed? Have they accounted for the possible presence of serial correlation in the series?

*It is now clarified:*

- *The duration is computed as the sum of all months from the onset with negative values, the peak is the month in which the episodes reach the highest value of SPEI1, and the severity is calculated as the sum of all SPEI values (in absolute values) during the episode.*

*Besides, the whole section 2 was improved.*
*In the case of the trend we just made a regression and the significance is calculated using the Wald test with t-distribution of the test statistic, and the trend is considered significant when p < 0.05. In this new version the results were similar after utilize the Mann Kendall test and the Sen´s slope.*

*At first we didn't consider the presence of serial correlation in the series. I order to assess the possible autocorrelation of the series was utilised the function of autocorrelation (ACF) in the R program. For the SPEI1 the autocorrelogram show, as expected, non-significant values. However, for some WTs (e.g. NE) (see right panel below) it seems that exist a clear seasonality and several values are significant, confirming the autocorrelation. To solve this concern we applied the Mann-Kendall Test of Prewhitened Time Series Data in Presence of Serial Correlation Using the von Storch (1995)*

*Approach. A detailed description of this method was added to section 2. (next figures are just an example and won't be included into the manuscript).*

[Figure]

The authors should clarify the rationale behind constraining their study from 1980, while EOBS dataset extends back to the 1950s (probably not for all climate variables).

*Despite the EOBS data is available from 1950s other variables (such as Era-Interim data used for WTs computations) is from 1979 and at first moment the river discharge data were from 1980. Thus, we decided to reduce the study to the period 1980-2017.*

I would recommend adding a new figure, in which the authors compare the accumulated SPEI values corresponding to each WT.

*Here we follow your instruction: Figure 6: I would recommend plotting the events at the xaxis, while the stacked bars show the contribution of each WT to such events. This will deliver the message clearer.*

*So, the new figure is compound by 10 new figures (10 episodes) where are shown the SPEI1 and each WT anomaly. A similar figure could be accumulating the anomalies of each WT, but the first choice you recommend also illustrate better the real frequencies of each WTs highlighting the percentages at the SPEI peak, or at the end of the episode. Besides, on a single month several WTs occurs and we have the percentage of days for each WT every month, while just one SPEI value for each month, because it is calculated at monthly scale, so, it is not possible to accumulate the SPEI values corresponding to each WT.*

Prior to calculating correlation, it is important to detrend the series of the frequency of WT.

*The series of WTs and SPEI were de-trended and the correlation was computed again. In this case the spatial correlation was also made.*

In the discussion, the authors should refer to the role of zonal and meridional circulation in drought characteristics (in the context of the directional WTs).

*We improved the discussion considering this advice.*

It is recommended to add a table, which summarizes trends of the 10 main WTs over the study period, their statistical significance and compares those with the trends observed for the different drought metrics (e.g. drought, severity, and occurrence).

*We calculated the trend for each WT and also for individual months. We applied the Mann-Kendall Test of Prewhitened Time Series Data in Presence of Serial Correlation Using the von Storch (1995) Approach. As all p values are greater than 0.05, trends are not statistically significant.*

|  | NE | E | SE | S | SW | W | NW | N | C | A |
|---|---|---|---|---|---|---|---|---|---|---|
| Z-value | 1.04 | 1.43 | -0.29 | 4.11 | 0.64 | -0.37 | -0.22 | -0.18 | -0.16 | -0.70 |
| Slope | 0.002 | 0.004 | -0.00 | 0.000 | 0.000 | -0.0002 | -0.0002 | -0.0001 | -0.0003 | -0.004 |
| *p*-values | 0.29 | 0.15 | 0.77 | 0.96 | 0.52 | 0.70 | 0.81 | 0.85 | 0.86 | 0.48 |

*Also, no statistically significant trends appear in almost every WTs and months. Significant trends only appear for SE in February, S in March, SW in September, C in August and A in April. Moreover, in most of the cases the trend show low values. We did not added the table above into the manuscript, but commented the results.*

[Figure]

HS method for calculating PET is a temperature-based method, which is more suitable for arid and semi-arid regions (not humid regions like the study domain). Have the authors tested the performance/accuracy of this method in their region? Several studies reported a less performance of temperature-based methods in assessing PET in humid climates.

*We agree about your concern. The limitation to computed the Eto thought the Penman Monthie method lies on the lack of several variables at high resolution for a long study period. To support our choice, the next paragraph was added to the manuscript.*

*By this method we do not consider that relative humidity and wind speed are also important factors that determine the vapour density above the soil surface and the aerodynamic resistance for vapor transport, permitting more realistic Eto values and consequently drought assessment (Bittelli et al., 2008; Vicente-Serrano et al., 2010; WMO, 2012; Davarzani et al., 2014). However, even though the Penman-Monteith offers a more accurate estimation of reference Eto than the Hargreaves formula (Tomas-Burguera et al., 2017), results of Vicente-Serrano et al., 2014 showed that Eto in Spain*

*estimated by the Hagreaves-Samani method for the period 1961 – 2011, had the closest agreement with the Eto obtained by the Peaman Monthie method in terms of temporal evolution and magnitude respect other eleven methods. These authors also found high correlations between Eto obtained by both methods in the northwest IP.*

*Vicente-Serrano et al., (2014). Reference evapotranspiration variability and trends in Spain, 1961–2011. Global and Planetary Change, Volume 121, October 2014, Pages 26-40. https://doi.org/10.1016/j.gloplacha.2014.06.005*

Given the limited area of the study domain, the spatial resolution of SMroot data seems to be coarse (0.25° × 0.25°) to provide a reliable assessment of the response of soil moisture to precipitation deficit. Also, in humid climates like those of the study area, the response of soil moisture to accumulated precipitation deficit is more pronounced at longer time scales (not 1-month time scale). The persistence of negative soil moisture anomalies is expected to be higher when there is a cumulative long-term decrease in the amount of precipitation. This aspect should be discussed thoroughly.

*In order to perform a better analysis, the figure and explanation for this section was removed. We considered your concern regarding the low resolution of the SMroot, which is representative for around 10 grid points. A similar analysis was performed but utilizing the soil moisture from the high-resolution (~4km) global dataset of monthly climate and climatic water balance "Terraclimate". We also utilized the Standardized Soil Moisture Index (SSMI) (Haoand AghaKouchak, 2013)*

Describe all symbols given in Eq. 1.

*Symbols are now described:*

*we used the method proposed by Hargreaves and Samani (1985) based on temperature data to estimate the Eto according to Equation 2:*

$$Eto=0.408*Ch*Ra* (\sqrt{(Tx-Tn)})+(Tm+17.8) \qquad (2)$$

*Where Ch = 0.0023; Ra is the extraterrestrial radiation (derived from the latitude and the month of the year), and Tx, Tn and Tm the maximum, minimum and mean temperature respectively.*

Which index exactly of the NAO, as well as ENSO, was used? Please, be more specific. There are different indices for quantifying each of them.

*We added this and more information.*

*To obtain the Northern Hemisphere teleconnection indices utilized in the study the Climate Prediction Center (CPC) utilise the Rotated Principal Component Analysis (RPCA) method used by Barnston and Livezey (1987) but utilizing monthly mean standardized 500-mb height anomalies obtained from the CDAS in the analysis region 20°N-90°N. This procedure isolates the primary teleconnection patterns for all months and obtain the index time series.*

*Barnston and Livezey (1987, Mon. Wea. Rev., 115, 1083-1126)*

Section 2.5 should be placed earlier in the materials and methods section (before the description of drought calculation).

*The section was moved*

The role of aerodynamic components in drought evolution should be discussed, given that these influences are not considered in HS method.

*The next paragraph was added to section 2 in order to expose this topic, and along the manuscript are discussed other issues:*

*By this method (HS) we do not considered that relative humidity and wind speed are also important factors that determine the vapour density above the soil surface and the aerodynamic resistance for vapor transport, permitting to obtain more realistic Eto values and consequently drought assessment (Bittelli et al., 2008; Vicente-Serrano et al., 2010; WMO, 2012; Davarzani et al., 2014). However, some studies have shown similar Eto estimations by means of the Penman-Monteith and Hargreaves methods in Spain (López-Urrea et al. 2006; Gavilán et al. 2008; Vanderlinden et al. 2008; López-Moreno et al. 2009), although the Penman-Monteith offers a more accurate estimation of reference Eto than the Hargreaves formula (Tomas-Burguera et al., 2017).*

II.    Minor comments

Title: It is recommended to indicate the location of the study domain (i.e. NW Iberia), as the majority of the NHESS readers are not familiar with the study basin. Also, it is important to include "hydrological droughts" in the title.

*The title was changed as follow: ''Hydrometeorological droughts in the Miño-Limia-Sil hydrographic demarcation (NW Iberian Peninsula): The role of atmospheric drivers''*

*In order to emphasize more on the hydrological issue, two figures showing the temporal evolution of the Standardized Soil Moisture Index (SSMI) (Haoand AghaKouchak, 2013) and the Standardized Runoff Index (SRI)(Shukla and Wood, 2008) were added. A detailed explanation of both index was added in section 2.*

P1 - L14 and other parts of the ms: "period of" <> "period".

*Thank you. It has been changed along the text*

L16 and other parts of the ms: "mo" <> "month".

*It was changed along the text*

L17: For a study that covers 38 years, the use of the term "historically" is misleading; please, define the confidence interval at which the significance was assessed.

*The historically term was removed from the text and the confidence level was provided (95%).*

L18: "different" <> "the different".

*Changed*

L19: Based on which scheme this classification was made? The abstract should stand alone based on this basic information.

*It was included*

- *... a daily weather type classification based on standard Lamb scheme was utilised for the entire Iberian Peninsula.*

L20: "were directly related to dry and wet conditions" This statement is vague, with no clear phrasing. It does not make a clear conclusion on whether these weather types are favoring for above-normal or below-normal precipitation.

*This sentence was removed and rewritten as follow:*

*Frequency and correlation analysis show that weather types conditioning an atmospheric circulation from the south-east/west, east/ west, north-east/west and the pure anticyclonic/cyclonic are associated to dry/wet conditions in the Miño-Limia-Sil Hydrographic Demarcation.*

L25: Please, define the rainy season.

*It was now defined: (October – May)*

L27 and other parts of the ms: "1 y" <> "1 year".

*It was changed along the text*

It is unclear how meteorological droughts assessed at 1-month time scale can be linked with land use changes (which almost occur at a coarse temporal scale".

*You are right, it was our mistake because at the same time were writing of a similar manuscript with a different scope. The sentence was removed.*

P2 - L5: Please, give some examples of these thermodynamic factors (e.g. wind speed, air pressure).

*In this case we modified the paragraph. Besides, in section 2 are explained the role of humidity and wind on the modulation of the Evapotranspiration and consequently the occurrence of dry conditions.*

- *This phenomenon is usually considered a prolonged dry period in the natural climate cycle that can occur anywhere in the world. It is initially caused by a lack of rainfall as well as for thermodynamics processes that affect content of water on the soil, which are induced by the wind speed, temperature, relative humidity and solar and long-wave radiation (WMO & GWP, 2016; Vicente-Serrrano et al., 2010; Sereviratne, 2012; Miralles et al., 2019).*

- L15: Delete "e.g." –

*It was deleted*

L19: "the precipitation" <> "precipitation".

*Changed*

L25: "land" Do you mean air temperature? LST has a different conception and is mostly assessed using remote sensing products (e.g. MODIS, AVHRR), which are only available for the most recent decades.

*The reviewer is right, it was clarified*

L28: The study of Vicente-Serrano et al. (2011) does not provide any assessment of future projections of precipitation.

*You are right. The reference was removed and added the correct.*

*Spinoni, J., Vogt, J.V., Naumann, G., Barbosa, P. and Dosio, A. (2018), Will drought events become more frequent and severe in Europe?. Int. J. Climatol, 38: 1718-1736. doi:10.1002/joc.5291*

P3 - L8: "a homogeneous region in terms of the total P variance over the IP". This statement should be elaborated thoroughly.

*This argument has been better discussed in the analysis of EOF of the SPEI.*

P4 - L25: "A drought episode was considered to occur when the SPEI at the temporal scale of 1 mo fell below zero, reached a value of at least -0.84, and later returned to positive values". This definition should be made simpler.

*The episode characterization was rewritten.*

*A drought episode occurs everytime the SPEI1 is continuously negative and reaches the value of −0.84 or less.*

P5 - L1: "Results" <> "Results aepisond discussion".

*Changed*

L9: Language and style should be revised.

*Thank you. We improved it.*

L12: "modulate" <> "impact".

*Changed*

It is unclear why the classification of weather types is only restricted to the period 1989-2017.

*This period is not correct, we apologize. The correct is 1980-2017. I was corrected in the text. This period was selected based on the available ERA Interim reanalysis data together with the rest of datasets in the moment we began to do this research (in particular the river discharge availability).*

L15: Please, define this spatial window.

*It was defined*

For classifying weather types, the authors should clarify how SF, WF, ZS, ZW, F, and Z were computed?

*The complete section was improved and the formulas were added:*

$SF = 1.305[0.25(p_5 + 2 \times p_9 + p_{13}) − 0.25(p_4 + 2 \times p_8 + p_{12})]$

$WF = [0.5(p_{12} + p_{13}) − 0.5(p_4 + p_5)]$

$ZS = 0.85 \times [0.25(p_6 + 2 \times p_{10} + p_{14}) − 0.25(p_5 + 2 \times p_9 + p_{13}) − 0.25 \times (p_4 + 2 \times p_8 + p_{12})$

$$+0.25(p_3 + 2 \times p_7 + p_{11})]$$

$$ZW = 1.12 \times [0.5 \times (p_{15} + p_{16}) - 0.5 \times (p_8 + p_9)] - 0.91 \times [0.5 \times (p_8 + p_9) - 0.5 \times (p_1 + p_2)]$$

$$F = (SF^2 + WF^2)^{1/2}$$

$$Z = ZS + ZW$$

P6 - L20: "from daily values" <> "aggregated from daily values".

*Modified*

L21: The name of the station "Albufeira Do Alto" does not fit with that labeled in Figure 1.

*This Figure has been modified.*

P7 - L5: "the annual cycle" <> "the year"; "western" <> "the western".

*Changed*

L20: What is the difference between "extensive" and "intense"? Do you refer to drought duration and severity?

*Yes, it was clarified into the text.*

P8 - L30: "for in" <> "for".

*This expression does not appear in P8, it was changed in P18.*

The acronyms "WTs" and "CWTs" are used interchangeably er in the text. - P23 (L10): "the soil" <> "soil".

*It was now corrected and only WTs is now used in the text*

P23 (L10): "the moisture" <> "moisture".

*Changed*

This work emphasized that drought did not respond linearly to most of the dominant circulation patterns in this region (apart from SCAN, AO) at 1-month timescale (Figure 10a). This finding should be discussed thoroughly in the text and linked with available literature.

*We agree and the explanation was improved.*

Tables

Table 1: There is a refinement of the drought categories of Agnew (mild drought is masked with another category).

*Modified!*

*Table 2: Trends in SPEI values should be expressed in z-units/year.*

*Done*

Figures

Figure 1: In the legend and caption, "rivers" <> "streams". The negative symbol corresponding to the longitudes should be deleted, given that the direction "W" is already included. It is important to include a distribution of the meteorological stations whose data were used for SPEI calculation.

*This figure was totally modified in order to follow your suggestions but also for representing the grid points utilized to compute the WTs.*

Figure 3: how were drought episodes defined? Have you applied n consecutive months with SPEI <-0.84?

*Yes. The explanation was improved in section 2. An episode was defined as the consecutive negative SPEI1 that reach at least the values of -0.84 or less.*

Figure 4: I would recommend using the anomalies (not the actual values) of SLP corresponding to the different WTs. This will facilitate defining the positive and negative centers of action that control air advection at the surface.

*Ok. Anomalies were calculated and will be added to the manuscript.*

Figure 5: The use of the symbol "x" should be described. The use of the legend in a vertical form is confusing, given that all WTs at the top of the panel show a negative correlation (shown in blue). I would recommend reversing the legend so that negative values of correlation are shown at the top, while negative correlations are illustrated below. Why the authors did not use a portrait diagram showing the interpolated surface of the correlation coefficient, with some contour lines to show the significance of the correlation? This will facilitate the readability of the figure.

*The X values indicate those correlations no statistically significant at p < 0.05. We reversed the legend. We considered to contour the correlations following your instructions, but, the function contour automatically interpolates and in this analysis the correlations are made for different WTs, so, in order to show the more precisely, we prefer to keep this format of circles.*

*A new figure showing the spatial correlations of the SPEI with each WT was made. This is NOT the last version of this figure, we will show just the demarcation, change the color and limits of the color bar and add a line for indicate the statistically significant values.*

[Figure]

Figure 6: I would recommend plotting the events at the xaxis, while the stacked bars show the contribution of each WT to such events. This will deliver the message clearer.

*We change this figure following your advice.*

Figure 7: why the percentages are given in negative? To which WT refers the "red" color? I would recommend adding a column to the three drought categories, which refers to wet conditions (i.e. SPEI values >0). This contrast can show interesting results about the role of each WT during dry vs. wet conditions

*This figure was improved following your suggestions. The red color don´t refer to any WT; it indicates the number of months under each drought category. The same figure was added for the wet conditions.*

---

## Author Response (AR2)

*Deadr Editor and Reviewer,*

*Please, consider our revised version of the manuscript "Hydrometeorological droughts in the Miño-Limia-Sil hydrographic demarcation (NW Iberian Peninsula): The role of atmospheric drivers" submitted by Sori et al. Thank you for your comments and advices in order to improve the manuscript.*

I. Major comments:

- In the Introduction, the authors should highlight the novelty/originality of their work by showing how their work is different from earlier studies conducting for the same study domain (e.g. Russo et al., 2015).

*The following paragraph has been added to the Introduction in order address this issue.*

- *To the best of our knowledge, there are few published studies about drought considering the MLSHD as a whole. Ojeda et al., (2019) investigated the temporal evolution of agricultural and hydrological droughts in the major river basins of the IP during the period 1980–2014 through modelling datasets. Considering the importance of the MLSHD as a hydrological unit, our main aim is to investigate the meteorological droughts which have affected the MLSHD using high resolution gridded datasets, but also to evaluate the role of atmospheric circulation and large-scale teleconnection patterns in the occurrence and magnitude of droughts over a longer period (1980–2017). Russo et al. (2015) carried out a similar analysis for the entire IP, but their approach identified the seasonal conditions of the drought state and related them to the frequency of weather types during 1950–2012.*

- The manuscript should be polished by a professional editor. It is not only the linguistic level but also the style. The text should be carefully edited to avoid repeated and trivial statements and reorganize several paragraphs.

*This new version has been revised by an English professional editor.*

- Have you tested that precipitation data in the study domain follow a log-logistic distribution? Fitting the SPEI to different statistical distributions can induce differences in SPEI values, especially extreme values (i.e. above 2 or below -2). I mean that there is always a degree of uncertainty in the calculation of the SPEI due to the effects of probability distributions and parameter errors.

*No, we did not test if the best probabilistic distribution to fit D series (P – PET) is the log-logistic. We followed the recommendation of the index developers and the generalized use of this distribution to calculate the SPEI over the Iberian Peninsula. But you are right, different probabilistic distribution may give mainly different extreme values of the SPEI. In the section of Methods has been added the next sentence in order to clarify this issue.*

- *Previous studies have been also used the log-logistic distribution to obtain the SPEI series for the IP (e.g. Russo et al., 2015; Páscoa et al., 2017; Coll et al., 2017; Ojeda et al., 2019). For our study region, it is possible that the D values demonstrate a better fit to a different probabilistic distribution; however, this can also occur for different accumulation periods of D (Monish and Rihana, 2020). The use of different probabilistic distributions to fit the D series may primarily affect the tail of each distribution and the extreme SPEI (Vicente-Serrano et al., 2010; Vicente-Serrano and Beguería, 2016), e.g. the [−2.33, 2.33] bounds (1 event in 100 cases). For these reasons, we preferred to use the distribution suggested by the authors of the index to calculate SPEI on a time scale from 1 to 24 months. For the calculation of the SPEI the R package available at http://cran.r- project.org/web/packages/SPEI is utilised. It includes all the recommendations proposed by Beguería et al., (2014).*

-

The authors adopted a daily weather type's classification similar to earlier studies in the IP (e.g. Ramos et al., 2014), even the selected 16 MSLP points are typically the same and accordingly the frequency of obtained types is expected to be identical. As such, the size of the manuscript can be reduced by summarizing the methodological part related to weather type's classification and referring mainly to the article by Ramos et al. 2014 or other similar studies, which described this method.

*We understand and we agree with you, so removed some sentences. However, another reviewer suggested to expand and explain well this methodology and show the equations. So, we tried to resume something but keeping the formulas.*

The rationale behind the computation of SPEI at a 1-month timescale to define the meteorological drought is needed. Why the authors did not consider 3 or 6-month timescales?

*In the Introduction there is a sentence that answer part of this question.*

- *In this study we focus on meteorological droughts attempting to explain the primary cause of other types of droughts (e.g. agricultural, hydrological, socioeconomic), which are normally associated with the reduction of the soil moisture content, low river flows, low water levels in rivers, lakes and groundwater, and socioeconomic impacts because the water scarcity (WMO, 2012). Investigating drought characteristics and generating mechanisms is essential to improve the forecasting methods and to take actions such as monitoring and early warnings, which contribute to efficient water management and reduction of the ecosystems and social vulnerability. We expect that our results will contribute to increase the hydroclimate knowledge of the MLSHD, support early warning, and strength drought management plans.*

*We added this sentence in the methodology.*

- *We focus on the SPEI at one-month temporal scale (SPEI1) to identify meteorological drought episodes. At this time scale the SPEI, as the SPI, can reflects short-term conditions, and consequently its application can be closely related to meteorological types of droughts (WMO, 2012).*

*We added a new paragraph in the Results.*

- *Figure 5a shows the temporal evolution of the SPEI1 computed for the MLSHD during 1980–2017. Drought conditions are observed in periods such as 1989–1992, 2004–2007, and 2015–2017, in agreement with results obtained by other authors for the NWIP through different indices (e.g. Garcia-Herrera, 2007; Andrade and Pereira, 2015; Spinioni et al., 2016; Ojeda et al., 2019). At this time scale, the negative values of the SPEI were primarily related to meteorological drought, which was unable to diagnose the agricultural, hydrological, and socioeconomic types of drought that are typically associated with the SPEI at greater temporal scales (WMO, 2012). However, meteorological droughts can be perceived as the initial cause of further types of droughts, since these are triggered in this case by the deficit of P combined with high temperatures and significant Eto. The identification of meteorological drought episodes affecting the IP has been a topic of research during recent years (e.g. Lana et al., 2006; Lorenzo-La Cruz et al., 2013; Páscoa et al., 2017; González-Hidalgo et al., 2018). As a drought episode was considered to occur every time the SPEI1 was continuously negative and reached the value of −0.84 or less; this threshold was identified by the black dashed line in Figure 5a. The onset, termination, and duration of these episodes are shown in Figure 5b.*

- I have also a concern regarding the use of the E-OBS gridded dataset, mainly related to the development and processing procedures of this dataset and thus the uncertainty in precipitation data. According to Hofstra et al. (2009), this product suffers from temporal inhomogeneities in some records. Also, the density of stations in this product varies considerably over time. Have the authors validated the performance of this product against observational data in their domain? Overall, I think it is important to discuss the limitations of this product.

*We did not validate this against observational data taking into account that the source of EOBs is the observational data and at first we did not know these limitations. But you are right. We searched for some publications to discuss about the accuracy of EOBS datasets. The next paragraph has been added to the manuscript.*

- *Monthly gridded data of precipitation, maximum temperature, and minimum temperature were obtained from daily values of the E-OBS v.18e gridded dataset (Cornes et al., 2018) with a longitudinal and latitudinal resolution of 0.1°*

*for the period 1980–2017. Owing to their high resolution, these datasets have been utilised to investigate extreme precipitation (Tabari and Willems, 2018) and drought events in Europe (Manning et al., 2019). However, the sparse distribution network in some European regions has led to an over-smoothing of precipitation intensities (Hofstra et al., 2009, 2010; Sunyer et al., 2013; Herrera et al., 2019). A comparison between daily precipitation and temperatures from standard and ensemble EOBS datasets with the observational gridded dataset Iberia01 performed by Herrera et al. (2019) revealed the main differences of temperatures occurred in the south, around the Guadalquivir and Guadiana basins, and respect the precipitation the high biases in the central IP and the Mediterranean regions. In addition to the high resolution, these datasets were chosen for this study because they provide both precipitation and temperature fields necessary for the computation of the Standardised Precipitation-Evapotranspiration Index (SPEI), thus minimizing errors that could arise due to the mixing of different data sets.*

- I am wondering why the authors included two indices (AO and NAO) whose phases are mostly similar.

You are right, there is a wide agreement on the similarity of both patterns on the north Atlantic region. Despite the strong impact of NAO on the Iberian and Europe precipitation occur in winter months, we correlate the series of all months separately. Some findings found a different trend between both index, so, we decided to use the AO too. The next paragraph has been added in the manuscript order to explain why we used both.

- *According to Wanner et al. (2001), the AO is similar to the NAO in many aspects. Multiple results have shown a strong relationship between the winter tropospheric pattern of the NAO and AO (Wanner et al., 2001; Rogers and McHugh, 2002; Hurrel, 2003; Dai and Tan, 2017). However, although the AO is strongly correlated with the NAO, it does not show the recent sustained significant summer decrease, but it does show enhanced early winter variability (Hanna et al., 2015), the active phase, as previously identified by Zhou et al., (2001). Results of Tabari and Willems 2018 show that the AO signal is oppositely related to anomalies of daily precipitation extremes during summer in NWIP, a phenomenon they did not see occurring with the NAO. Therefore, in order to assess any possible difference in the impacts over drought conditions, both indexes were used in this investigation. The negative phase of the NAO was associated with the weakness of the Azores High, and a southwards position in the storm tracks, thereby resulting in wet conditions over the IP (Trigo et al., 2002). The correlations in Figure 15a demonstrate that both the NAO and AO relationships with the SPEI were nearly identical across different temporal scales. However, the correlations with the AO are greater, indicating that the AO index may be more effective for explaining the atmospheric influence on dry/wet conditions in the MLSHD. Nevertheless, the NAO index has also been traditionally defined as the normalized*

*pressure difference between a station on the Azores and one on Iceland (Hurrell, 1995; Jones et al., 1997); therefore, the correlations with the SPEI could be also different in this regard.*

- I recommend adding a figure that shows the temporal variability of SPEI corresponding to the different EOFs.

*We tried to make this plot but was difficult to follow a line over a line. We even plotted the SPEI in bars and the PC1 in line; and a second figure showing the PC2 and PC3 both as line. The results were not clear to follow. Thus, we decided to add just the figure of the PC1,2 and 3 for figure 4 (originally Figure 3). These correspond to the whole period of the study.*

*We also showed the EOF 1 to 6 for those months when the average SPEI was under -0.84. For this case we did not plot the 6 PCs. The reason is that this EOF are calculated for a period of months that is not continuous, so, we plotted the PCs of the EOF calculated to all the period of the study (as above mentioned).*

- In the results and discussion, the topical sentences of many paragraphs are misleading.

*Thank you, it has been improved.*

- There is no need to use abbreviations for precipitation and maximum and minimum air temperatures.

*We changed this along the manuscript. However, grammar editor suggested to add abbreviations for: northwest IP – NWIP.*

**II. Specific comments:**

- L15: add the abbreviation (SPEI) and avoid the use of the full name of the index in the subsequent lines.

*Added.*

- In the abstract, it is not clear at which timescale the SPEI was computed?

*We use SPEI1 to calculate drought and added this information in the abstract.*

- L16: "December" <> "and December".

*It was changed.*

- L22: I suppose you refer to the "hydrological year".

*Yes, it was changed.*

- L28: Define the "rainy season" and specify which timescales do you mean?

*Defined and add in the sentence:*

- *"Hydrological drought investigated through the Standardised Runoff Index was closely related to dry/wet conditions revealed by the SPEI at shorter temporal scales (one, two months), especially during the rainy months (December-April)."*

P2:

- L5-6: You should revise this statement clearly to distinguish between the thermal and aerodynamic forces of drought.

*The sentence was rewritten as follows:*

- *It is initially caused by a lack of rainfall as well as for thermodynamics processes (e.g. turbulent fluxes and water phase transitions) (Wehrli et al., 2018) induced by aerodynamics (wind speed), radiative ant thermal factors (e.g. solar and long-wave radiation, high temperatures) (WMO & GWP, 2016; Vicente-Serrrano et al., 2010; Sereviratne, 2012; Miralles et al., 2019).*

- L8-9: This statement should be elaborated thoroughly in the text.

*To do it we added a new figure showing the temporal evolution of the SPEI from 1 to 24 temporal scales.*

- L13-15: This should be revised in the context that other studies –based on long-term data- showed non-significant trends in drought (e.g. Lloyd-Hughes and Saunders, 2002; Spinoni et al., 2017, 2019).

Thank you. We improved this.

- *Concerning the existence of the trends in droughts, Lloyd-Hughes and Saunders (2002) found a significant negative linear trend on the series of the Palmer Drought Severity Index (PDSI) over the period 1901–1999 in the northwestern IP (hereafter; NWIP). Higher atmospheric evaporative demand increased the severity of climatic droughts during the period of 1961–2011 in the IP, which contributed to a decrease in surface water resources (Vicente-Serrano et al., 2014). However, these authors also argued that drought variability has mainly been controlled by precipitation. Coll et al. (2017) reported that the rise in temperature was responsible for the greater drought severity and larger surface area affected in the IP from the 1980s to 2010 (with respect to the period of 1906–2010), as it led to an increase in atmospheric evaporative demand. A significant tendency towards dryness during 1975–2012 in the IP was also revealed by Páscoa et al. (2017). These authors showed that the northwestern region of the IP was particularly affected by these trends. Over a shorter study period (1974–2010), Gómez-Gesteira et al. (2011) found a significant increasing trend of 0.5 °C per decade in air temperature in this same region and 0.24 °C per decade in the sea surface temperatures of the adjacent Atlantic Ocean, but annual precipitation did not show any significant trend in the interior, which suggested a possible dominant role of temperature on the occurrence of drought in this region. Although the results described above agree with respect to the occurrence of a trend towards drier conditions in the IP during recent decades, findings of Spinioni et al. (2017) reveal noticeable differences with respect to the trends and severity of droughts among the winters of 1950–2014 and 1981–2014, and the spring, summer, and autumn*

*seasons of 1950–2015 and 1981–2015 in the IP. Overall, the high confidence level that global warming is likely to reach 1.5 °C above preindustrial levels within a short period (between 2030 and 2052) if the current rate of increase continues is presently a serious concern (IPCC, 2018). In this sense, the IP is considered as one of the most likely European regions to suffer from an increase in drought severity during the 21st century (Spinioni et al., 2018). However, Trenberth et al. (2014) argued that increased heating from global warming may not necessarily cause more droughts, but when they do occur, would be expected to exhibit rapid onset and greater intensity.*

- L15: Do you refer to "air temperature" or "land surface temperature"?

*The correct is air temperature. The sentence was modified to clarify this.*

- L16: "any trend" <> "any significant trend".

*Changed*

- L19: "one" <> "as one"; "suffer" <> "suffer from".

*changed*

- L24-25: Language should be revised.

*Revised.*

P3

- L13: a reference is needed.

*Added*

*Vicente Davila, F.., Méndez-Martínez, G., and Fidélis, T.: Transboundary strategic environmental assessment in river basins: the case of Minho. MONFRAGÜE DESARROLLO RESILIENTE, ISSN 2340-5457, N° 2. https://www.researchgate.net/publication/274640892_Evaluacion_ambiental_estrategica_transfronteriza_de_cuencas_hidrograficas_la_experiencia_del_Mino, 2015.*

- L14: "not any" <> "no".

*Changed*

- L22: Revise the numbering.

Ok!

- L23: "20 000" <> "20,000".

*Changed*

- L25: add the separator "," to the number.

*Added*

- L27: "said"??

*The word has been deleted.*

P4

- Figure 1: The panel belonging to the study area should be enlarged to show the topographical gradient and main sub-basins. The 16 points can be shown in an inset map.

*We enlarged the image of the basin, because is difficult to make an inset map showing the points from a figure of the basin. Now is possible to see better the basin.*

- L19: "24° C" this belongs to maximum air temperature?

*Thank you for the comment. We changed this sentence and put : "the maximum mean temperature occurred in July and August (26°C).*

- L20: "regular" <> "regulated".

*Changed*

P5

- Figure 2: The description of the boxplot and the whiskers should be revised carefully.

*The explanation was revised; you can read it below.*

- L8-11: Please, define the climatic variables derived from E-OBS.

*Defined*

P6

- L26-28: Some citations of these works are needed.

*Added*

*In consequence, it has been applied to a large variety of ecosystems across the world for identifying dry and wet conditions and evaluating drought recurrence (Potop et al., 2013; Vicente-Serrano et al., 2016; Salah et al., 2017; Sordo-Ward et al., 2017; Wang et al., 2018).*

P8

- L6: "the sum" <> "the maximum length"; "negative values" <> "negative values less than -0.84".

*Changed*

- L15: "eigenvectors" <> "eigenvalues".

Yes, thank you. ''eingevalues''

- L19: "Mann-Kendall" <> "modified Mann-Kendall"; how the magnitude of change was assessed? At which timescale the significance was defined?

*We improved this explanation as follows:*

- *Trend analysis using the Mann-Kendall Test (Mann, 1945; Kendall, 1975) of Prewhitened Time Series Data in Presence of Serial Correlation using the von Storch (1995) approach was performed. This approach ensures to avoid possible autocorrelation of the series. The null hypothesis (H₀) is that the data come from a population with independent realizations and are identically distributed, while the alternative hypothesis (Ha), is that the data follow a monotonic trend. With a significance level of 0.05 the null hypothesis of no trend is rejected if the Z score is greater or less than the critical value 1.96. The Sen's slope (Sen, 1968) is used to determine the magnitude of trend increasing or decreasing in the period of study. The combination of both methods is often used to analyze the trend change of hydrometeorological time series such as precipitation, runoff, drought index, etc. For this analysis the R package 'modifiedmk' (Patakamuri and O'Brien, 2019) is utilised.*

P9

- L5: "objective" <> "semi-objective".

*Changed*

- L8: "over the IP" <> "over the IP and surrounding regions".

*Changed*

- L12: "3-8" <> "4-9".

*Changed*

P10

- L5: Delete "daily".

*Deleted*

P11

- L1: How is this finding verified in this study?

*This sentence was removed because the text was rewritten and is not important to mention now. However, the first idea was to mention that as the SPEI1 is calculated taking into account the monthly balance of precipitation minus precipitation, it should be more related than on other time scales to the actual monthly balance of (E-P).*

- L7: The amount of variance of EOFs 2 and 3 is too low. I am wondering whether you applied the correlation or covariance matrix for the input data.

*- We applied the covariance method.*

- Figure 3: The plotted values seem to be very low!

*You are right. We made a mistake in one point because we standardized the series when the series was already standardized. We remove this, founding the same spatial EOF, but and even that the final values despite are higher, they are relatively low.*

- L25: This is not evident.

*The sentence was removed.*

P12

- Figure 4: add the label to y-axis.

*SPEI unit is non dimensional, just units.*

- L8: "on" <> "in".

*Changed*

- Table 2: I am wondering that some values of the slope is statistically significant, while their values are very low (e.g. SPEI1). Please, revise the units carefully. Also, why was the statistical significance assessed using t-test, not MK test?

*We mention now both. But we used modified prewhitene MK-Test. As commented in the methodology we utilized the modified MK-test in order to remove the possible autocorrelation of the series. Yes the values are very low. You can see in the caption below the results.*

[Figure]

And for the Duration and Severity.

```
> pwmk(datos1$Dur)
        Z-Value        Sen's Slope  old. Sen's Slope        P-value                S
    2.874530e+00      1.744667e-02      0.000000e+00    4.046296e-03      8.520000e+02
          Var(S)               Tau
    8.764467e+04      2.035356e-01
> pwmk(datos1$Sev)
        Z-Value        Sen's Slope  old. Sen's Slope        P-value                S
   -1.979834e+00     -2.376525e-03     -2.515541e-03    4.772215e-02     -5.880000e+02
          Var(S)               Tau
    8.790600e+04     -1.404682e-01
```

P13

- Table 3: "severe" <> "anomalous"; add the units to "severity" and "duration".

*The unit for duration has been added. Severuty, don't have unit, it is adimensional.*

- Figure 5: I am wondering that the P25 for some months correspond to 500-600 mm.

You are right, we apologize. It was a mistake in the legend. The correct percentiles are p05, p25 and p50. We changed it on the figure and the text.

P14

- L1: "similarity" <> "homogeneity".

*Changed*

- Figure 6: The plotted values are very low!

*We obtained similar values (and variances) applying either the correlation or the covariance method. At first we had the same concern and look for in other articles founding the similar values for the EOF of the SPEI.*

- L16: What do you mean by "removing the respective grand means"?

It was a mistake. The word ''grand'' was removed!

- L20-21: This description should be revised.

*We made a mistake with the number of the figure.*

*The reddish (blueish) isolines in Figure 7 identify the higher (lower) values in the MSLP absolute fields and the positive (negative) values of MSLP anomalies.*

P18:

- Figure 9: "symbols" <> "dots".

*Changed*

**You can find bellow a version with track changes. However, the clean version has been finally improved and reviewed the grammar by an expert.**

**Hydrometeorological droughts in the Miño-Limia-Sil hydrographic demarcation (NW Iberian Peninsula): The role of atmospheric drivers**

Rogert Sorí[1,2], Marta Vázquez[1,2,3], Milica Stojanovic[2,3], Raquel Nieto[1], Margarida Liberato[2,3], Luis

5    Gimeno[1],

[1]Environmental Physics Laboratory (EPhysLab), CIM-UVigo, Universidade de Vigo, Ourense, 32004 Spain

[2]Instituto Dom Luiz, Faculdade de Ciências da Universidade de Lisboa, Campo Grande, Portugal

[3]Escola de Ciências e Tecnologia, Universidade de Trás-os-Montes e Alto Douro, Vila Real, Portugal

10    *Correspondence to*: Rogert Sorí (rogert.sori@uvigo.es)

**Abstract.** Drought is one of the main natural hazards because of its environmental, economic, and social impacts. Therefore, monitoring and prediction for small regions, countries, or whole continents are challenging. In this work, the meteorological droughts affecting the Miño-Limia-Sil Hydrographic Demarcation in the northwestern Iberian Peninsula during the period 1980–2017 were identified. For this purpose, and to assess the combined effects of temperature and precipitation on drought

15    -conditions, the one-month Standardised Precipitation-Evapotranspiration Index (SPEI1) was utilised. Some of the most severe episodes occurred during June 2016 – January 2017, September 2011 – March 2012, and December 2014 – August 2015. An Empirical Orthogonal Function analysis for the SPEI series revealed that the spatial variability of the SPEI shows a great homogeneity in the region, and consequently the drought phenomenon behaves in the same way. Particular emphasis was given to investigating atmospheric circulation as a driver of different drought conditions. To this aim, a daily

20    weather type classification based on Lamb weather type (LWT) classification (Lamb, 1972) was utilised for the entire Iberian Peninsula. The results showed that atmospheric circulation from the southwest, west, and northwest are directly related to wet conditions in the Miño-Limia-Sil Hydrographic Demarcation during the entire  hydrological year. Contrastingly, weather types imposing atmospheric circulation from the northeast, east, and southeast are best associated with dry conditions. Anomalies of the integrated vertical flow of humidity and their divergence for the onset, peak, and termination of the ten most

25    severe drought episodes also confirmed these results. In this sense, the major atmospheric teleconnection patterns related to dry/wet conditions were the Arctic Oscillation, Scandinavian Pattern, and North Atlantic Oscillation. Hydrological drought investigated through the Standardised Runoff Index was closely related to dry/wet conditions revealed by the SPEI at shorter temporal scales (one, two months), especially during the rainDecember-April).

Introduction

Drought is one of the most dangerous natural phenomena in many regions worldwide, as it affects a wide range of environmental, economic, and social sectors (Wilhite, 2000; McMichael et al., 2011; Stankeet et al., 2013; Gerber and Mirzabaev, 2017; Liberato et al. 2017; Guerreiro et al., 2018). This phenomenon is usually considered a prolonged dry period

5    in the natural water cycle that can occur anywhere in the world. It is initially caused by a lack of rainfall as well as for thermodynamics processes (e.g.  turbulent fluxes and water phase transitions)) (Wehrli et al., 2018) induced by aerodynamics  (wind speed), radiative ant thermal forces, solar and long-wave radiation, high temperatures)) (WMO & GWP, 2016; Vicente-Serrrano et al., 2010; Sereviratne, 2012; Miralles et al., 2019).

Drought propagation is also due to natural and human drivers through multiples feedbacks (Van Loon et al., 2016). The Iberian Peninsula (IP) in the Euro-Atlantic and Mediterranean regions is a drought-prone area (Páscoa et al., 2017) that has been affected by well-known record-breaking droughts in 2004-2005 (García- Herrera et al., 2007), 2011-2012 (Trigo et al., 2013), and 2015-2016 (García-Herrera et al., 2019). Concerning the existence of the trends in droughts, higher atmospheric evaporative demand increased the severity of climatic droughts in the period 1961-2011 in the IP, which contributed to the decrease in surface water resources (Vicente-Serrano et al., 2014). Although, these authors also argued that drought variability has mainly been controlled by precipitation. The temperature increase has been also  responsible for greater drought severity and larger surface area affected in the IP from 1980s to 2010 (respect 1906-2010) due to the increase in atmospheric evaporative demand (Coll et al., 2017).  a significant tendency towards dryness during  1975–2012 in the IP was also revealed by (Páscoa et al., 2017). These authors showed that north west of the IP was particularly affected by these trends. Over a shorter study period (1974–2010), Gómez-Gesteira et al. (2011) found a significant increasing trend of 0.5 °C per decade in air temperature in the north western IP and 0.24ºC per decade   in the sea surface temperatures of the adjacent Atlantic Ocean,  but annual precipitation did not show any significant trend in the , which suggest a possible dominant role of temperature on the occurrence of drought in this region. Despite the fact that the results described above coincide in the occurrence of a trend towards drier conditions in the IP during recent decades, findings of Spinioni et al (2017) reveal noticeable differences respect the trends and severity of droughts among winter of 1950-2014, and 1981-2014, and spring, summer and autumn 
[revised manuscript text omitted]

---

## Author Response (AR3)

*Dear Editor and Reviewer,*

*Please, consider our revised version of the manuscript "Hydrometeorological droughts in the Miño-Limia-Sil hydrographic demarcation (NW Iberian Peninsula): The role of atmospheric drivers" submitted by Sori et al. We read the manuscript*

5 *carefully in order to check for any possible mistake. Thank you for your time and advices to improve the manuscript.*

- Add the labels "Z-unit" to the y-axes of Figures 4, 5, and 17.

*Done*

- Figures 4 and 7: Label the scale bar of EOFs as "loadings". The caption, indicate that the numbers between parentheses correspond to the amount of variance explained by each EOF.

10 *Done*

- Table 2: revise the caption carefully.

*Done. And here we need to clarify that we made a mistake in the last version submitted, when utilized an incorrect series of severity. We carefully revised these results and added the trend for the number of episodes.*

- Table 2: The unit of SPEI-1 should be event/year; the unit for severity should be expressed as Z-unit/year.

15 *Done*

- Figures 9 and 15: "non-statistically-significant" <> "statistically non-significant".

*Done*

- Figure 18: The position of months and numbers on the x- and y-axis should be placed correctly (E.G. what are the values corresponding to SPEI-24, OCt, or Sept).

20 *Thank you. The figure was replaced by this new one.*

[Figure]

…......

- I do recommend careful editing of the language and style of this work to improve the phrasing (e.g. some
25  statements are very long and should be shortened). In the results and discussion, introductory sentences of some
paragraphs are irrelevant to the content of the paragraph.

*We followed your advice to improve this mistakes. We basically shortened sentences and removed some irrelevant sentence.
However, it was already sent to language editing, and we took special care for not modifying so much now.*

[revised manuscript text omitted]